# Variable paralog expression underlies phenotype variation

**Raisa Bailon-Zambrano[1†], Juliana Sucharov[1†], Abigail Mumme-Monheit[1], Matthew Murry[1], Amanda Stenzel[1], Anthony T Pulvino[1], Jennyfer M Mitchell[1], Kathryn L Colborn[2], James T Nichols[1]\***

[1]Department of Craniofacial Biology, University of Colorado Anschutz Medical Campus, Aurora, United States; [2]Department of Surgery, University of Colorado Anschutz Medical Campus, Aurora, United States

**Abstract** Human faces are variable; we look different from one another. Craniofacial disorders further increase facial variation. To understand craniofacial variation and how it can be buffered, we analyzed the zebrafish *mef2ca* mutant. When this transcription factor encoding gene is mutated, zebrafish develop dramatically variable craniofacial phenotypes. Years of selective breeding for low and high penetrance of mutant phenotypes produced strains that are either resilient or sensitive to the *mef2ca* mutation. Here, we compared gene expression between these strains, which revealed that selective breeding enriched for high and low *mef2ca* paralog expression in the low- and high-penetrance strains, respectively. We found that *mef2ca* paralog expression is variable in unselected wild-type zebrafish, motivating the hypothesis that heritable variation in paralog expression underlies mutant phenotype severity and variation. In support, mutagenizing the *mef2ca* paralogs, *mef2aa*, *mef2b*, *mef2cb*, and *mef2d* demonstrated modular buffering by paralogs. Specifically, some paralogs buffer severity while others buffer variability. We present a novel, mechanistic model for phenotypic variation where variable, vestigial paralog expression buffers development. These studies are a major step forward in understanding the mechanisms of facial variation, including how some genetically resilient individuals can overcome a deleterious mutation.

**\*For correspondence:**
JAMES.NICHOLS@UCDENVER.
EDU

[†]These authors contributed
equally to this work

**Reviewing Editor:** Tatjana
Piotrowski, Stowers Institute for
Medical Research, United States

## Editor's evaluation

In this elegant genetic study, Bailon-Zambrano et al. draw on classical genetic concepts to address the clinically pertinent question of how genetic variants in the same gene can yield wildly different phenotypes in different individuals. Specifically, this work makes significant contributions to our understanding of how a mutant phenotype may be modified by the expression levels of paralogues of the mutant gene.

## Introduction

### Human faces are variable

Human craniofacial variation allows us – and even computers – to identify specific individuals. A recent study indicated that human facial structures are more variable than other human anatomical features and that evolution selected for increased variation in craniofacial-associated genomic regions (*Sheehan and Nachman, 2014*). While human genomic variation contributes to craniofacial phenotypic variation (*Liu et al., 2012*; *Paternoster et al., 2012*), facial variation persists among more genetically homogeneous populations such as Finnish people (*Sheehan and Nachman, 2014*) and even monozygotic twins (*Kohn, 1991*). Therefore, even among individuals with similar genomes,

craniofacial development may remain noisy or sensitive to minor fluctuations, compared with other developmental processes.

Phenotype severity and variation can be measured in different ways. Penetrance is the frequency of a phenotype associated with a genotype. Expressivity is different degrees of the same phenotype associated with a genotype. Here, we use both penetrance and average expressivity to measure phenotype severity. We also use the distribution in expressivity to measure phenotypic variation. Furthermore, there are two types of variation. Among-individual variation can be quantified by measuring differences between individuals. Within-individual variation is measured by quantifying departures from symmetry on left versus right sides of an individual known as fluctuating asymmetry (*Valen, 1962*). It is unknown if among- and within-individual variation are products of the same biological mechanisms (*Hallgrimsson et al., 2019*). There is evidence to support that these two types of variation are associated (*Breuker et al., 2006*; *Santos et al., 2005*); however, other studies indicate that they are independent (*Debat et al., 2009*; *Breno et al., 2011*). Further work is needed to resolve this question.

## Variability in human craniofacial disease

In the 1940s, Waddington observed that mutant organisms are often associated with increased phenotypic variation compared to wild types (*Waddington, 1942*), and more recent studies support this finding (*Fish, 2016*; *Hallgrímsson et al., 2006*; *Scharloo, 1991*; *Parsons et al., 2008*). Human genetic craniofacial disease phenotypes also appear more variable than normal human facial phenotypes (*Ward et al., 2000*). For example, facial clefting among monozygotic twins can be incompletely penetrant or unilateral (*Trotman et al., 1993*; *Young et al., 2021*), demonstrating both among- and within-individual variation. How the same genetic disease allele can have devastating consequences in some individuals, while other resilient individuals can overcome the deleterious mutation is not well understood (*Chen et al., 2016*).

One human genetic disease that presents variable craniofacial phenotypes is *MEF2C* haploinsufficiency syndrome. Patients heterozygous for mutations affecting the transcription factor encoding gene *MEF2C* show variable facial dysmorphologies (*Le Meur et al., 2010*; *Zweier et al., 2010*; *Gordon et al., 2018*; *Tonk et al., 2011*). To our knowledge, no *MEF2C* homozygous mutant patients have been identified, likely due to embryonic lethality. Additionally, craniofacial asymmetry is documented in some patients with this disorder (*Nowakowska et al., 2010*; *Novara et al., 2013*). Thus, both among- and within-individual variation are present in *MEF2C* haploinsufficiency syndrome patients. Numerous *MEF2C* mutant alleles cause this disorder. Therefore, a model system in which the same allele can be studied in many different individuals is needed.

## Variable buffering by gene family members might underlie among-individual variation

During evolution, whole genome duplications produced multiple *mef2* genes, or paralogs, in vertebrate genomes. Thus far, no *mef2* paralogs are associated with craniofacial development besides *mef2c*. When whole genome duplications occur, the most likely outcome is the loss of one of the duplicate genes through the accumulation of deleterious mutations and eventual nonfunctionalization (*Takahata and Maruyama, 1979*; *Watterson, 1983*). However, sometimes mutations occur in gene regulatory elements partitioning gene expression domains among the new duplicates. This preserves both copies and is called subfunctionalization (*Force et al., 1999*; *Ohno, 2013*). Subfunctionalized duplicates may experience only partial loss of expression subdomains (*Force et al., 1999*), retaining relics of their ancestral expression pattern even if the gene is no longer required for the original function.

Compensation, or the ability of paralogs or gene family members to make up for the loss of a gene due to an overlap in function, has long been recognized as a source of genetic robustness (*Wagner, 2000*; *Kirschner and Gerhart, 1998*; *Gu et al., 2003*; *Conant and Wagner, 2004*; *Kamath et al., 2003*). Robustness is defined as the ability of a biological system to overcome genetic or environmental perturbation (*Diss et al., 2014*; *de Visser et al., 2003*). Buffering lessens the impact of perturbation on a system. Although buffering by paralogs has long been proposed, whether paralogs retain buffering capacity after subfunctionalization has not been sufficiently addressed.

We hypothesize that vestigial expression can buffer against loss of another paralog and that variation in vestigial expression underlies phenotypic variation. This hypothesis is supported by work in systems ranging from yeast to human cells demonstrating that paralogs contribute to genetic robustness (*Diss et al., 2014*; *De Kegel and Ryan, 2019*; *Dandage and Landry, 2019*; *Li et al., 2010*; *Hsiao and Vitkup, 2008*; *Wagner, 2008*; *Dean et al., 2008*). However, whether paralog expression variation underlies phenotypic variation has not been directly tested. Finally, whether variation in paralogous compensation is subject to selection is unknown.

We previously found evidence to support this hypothesis in zebrafish. *mef2ca* single mutants produce dramatically variable craniofacial phenotypes (*Nichols et al., 2016*). *mef2cb* is the most closely related *mef2ca* paralog. However, *mef2cb* is not required for craniofacial development, and homozygous mutants are viable and indistinguishable from wild types (*Hinits et al., 2012*). However, we reported that ventral cartilage defects associated with *mef2ca* mutation become more severe when we removed a single functional copy of *mef2cb* from *mef2ca* homozygous mutants (*De Laurier et al., 2014*). Our vestigial buffering hypothesis predicts that although *mef2cb* is no longer overtly required for craniofacial development, remaining *mef2cb* expression in neural crest cells might partially substitute for *mef2ca* loss. There is further evidence from mouse mutants that *Mef2* paralogs can functionally substitute for one another (*Majidi et al., 2019*). Because mutations in none of the other four zebrafish paralogous *mef2* genes have been reported, their function is unknown. These paralogs have human orthologs, *MEF2A*, *MEF2B*, and *MEF2D*. None of these genes have been associated with human craniofacial development or disease.

## We developed a system for understanding craniofacial development and variability

One prominent phenotype in zebrafish *mef2ca* homozygous mutants is expansion of the opercle, a bone supporting the gill flap. This phenotype is remarkably variable (*De Laurier et al., 2014*; *Kimmel et al., 2003*); *mef2ca* homozygous mutants have many opercle shapes, and some even develop wild-type-looking opercle bones; the phenotype is incompletely penetrant (*Nichols et al., 2016*). With some genes, mutants encoding a premature termination codon (PTC) upregulate compensating genes through transcriptional adaptation, explaining why some mutant alleles do not produce a phenotype (*Rossi et al., 2015*; *El-Brolosy et al., 2019*). Transcriptional adaptation does not contribute to incomplete penetrance in our system (*Sucharov et al., 2019*). Specifically, we pharmacologically inhibited transcriptional adaptation (by inhibiting RNA decay) and did not observe changes in penetrance.

We demonstrated that the factors underlying opercle variation are heritable through selective breeding, which shifted the penetrance of the expanded bone phenotype to generate strains with consistently low or high penetrance of this phenotype (*Nichols et al., 2016*). Unlike traditional mutagenesis modifier screens (*St Johnston, 2002*; *Kile and Hilton, 2005*), our selective breeding paradigm likely enriched for standing genetic variation without the need for further mutagenesis. Thus, we likely bred naturally occurring genetic modifiers in our background to fixation. Similar approaches have proven successful in other systems (*Vu et al., 2015*; *Gasch et al., 2016*). *mef2ca* is pleiotropic, and although we only selected on opercle bone penetrance, nearly all *mef2ca*-associated phenotypes increased or decreased penetrance in the high- and low-penetrance strains, respectively (*Sucharov et al., 2019*). The only phenotype remaining fully (100%) penetrant in the low- and high-penetrance strains is a shortened symplectic cartilage, a linear, rod-shaped cartilage that functions as a jaw support structure. In unselected lines, *mef2ca*-associated phenotypes are only found in *mef2ca* homozygous mutants (*Miller et al., 2007*). However, in high-penetrance heterozygotes, we observed the shortened symplectic cartilage, suggesting this phenotype is highly sensitive to *mef2ca* loss (*Sucharov et al., 2019*). Penetrance is a binary measurement (a shortened symplectic is present or not), and we have not yet examined expressivity (to what extent a mutant symplectic is shortened). We do not know if the average length of the shortened symplectic or the distribution of symplectic length is different between low- and high-penetrance strains.

Here, we capitalize on the strengths of our zebrafish system to address the mechanisms underlying craniofacial variation. We quantified expressivity by taking linear measurements of the symplectic cartilage, allowing us to measure severity, among-individual variation, and within-individual variation in the zebrafish craniofacial skeleton. Combining these measurements with penetrance scoring, we compared phenotype severity and variation between selectively bred strains. We found that increased

severity is associated with increased variation. In the high-penetrance strain, we found a more severe symplectic phenotype, and more among-individual variation but not within-individual variation compared to the low-penetrance strain.

What factors were selected upon from the original genetic background to increase or decrease penetrance? Work in nematodes demonstrates that expression variation in otherwise redundant genes contributes to variable penetrance in mutants (*Raj et al., 2010*). Therefore, we hypothesized that one or more of the five zebrafish *mef2ca* paralogs might be redundant with *mef2ca* and ameliorate the phenotype in the low-penetrance line or reciprocally intensify the phenotype in the high-penetrance line. We compared the expression of the *mef2* paralogs between selectively bred strains and found that many of the paralogs are more highly expressed in the low-penetrance strain compared to the high-penetrance strain. Furthermore, we uncovered standing variation in paralog expression in unselected strains. To determine if the increased paralog expression we discovered in the low-penetrance strain buffers the *mef2ca* mutant phenotype, we mutagenized the *mef2* paralogs. Double mutant analyses indicate that the different paralogs modularly buffer different aspects of the pleiotropic *mef2ca* mutant phenotype: some affect severity, while others affect variation and some affect both. These findings demonstrate that heritable, variable paralog expression is a major factor affecting phenotype severity and among-individual phenotypic variation but does not contribute to within-individual variation.

## Results
### Craniofacial phenotypes are more variable in *mef2ca* mutants compared with wild types

The phenotypic variation often associated with human genetic diseases might be partially due to individuals inheriting different mutant alleles retaining different levels of functional activity, expression domains, and/or transcriptional levels. A strength of the zebrafish system is that variation can be studied in many different individuals with the same mutant allele. Wild-type *mef2ca* encodes an N-terminal MADS box (MCM1, agamous, deficiens, and SRF) and an adjacent MEF2 domain (*Chen et al., 2017*; *Figure 1A*). These highly conserved domains mediate dimerization, DNA binding, and co-factor interactions (*Potthoff and Olson, 2007*). The *mef2ca^b1086* mutant allele arose from a forward genetic screen and produces a PTC just downstream of the MADS box. This mutant allele variably displays phenotypes including ectopic bone near the opercle, interhyal joint fusion, dysmorphic ceratohyal, reduced Meckel's cartilage, jaw joint fusion, and shortened symplectic (*Figure 1B and C*). Skeletal preparations from two full-sibling individuals illustrate both among- and within-individual variation easily observed in *mef2ca* mutants (*Figure 1B*). Comparing the two mutant individuals (B3 and B4) demonstrates among-individual variation associated with the opercle bone phenotype; one individual (B3) has phenotypically wild-type opercles, while the other individual has bilateral mutant phenotypes (B4). Meanwhile, within-individual (left-right) symplectic cartilage phenotype variation is present in one of the *mef2ca^b1086* animals (B4); in this animal, the symplectic cartilage is longer on one side than the other.

To quantify the variation associated with *mef2ca* mutants, we first scored penetrance of the various *mef2ca* phenotypes in an unselected AB background (*Figure 1C*). Based on penetrance, the symplectic cartilage is the craniofacial structure most sensitive to *mef2ca* loss (*Sucharov et al., 2019*). Moreover, we found that symplectic length correlates with opercle bone area (*Figure 1—figure supplement 1*), suggesting the shortened symplectic phenotype is a good proxy for the entire craniofacial complex. We developed a quantitative phenotyping assay where the symplectic length can be rapidly measured in many animals (*Figure 1D*). We used the total linear symplectic length (left side plus right side) to examine expressivity (*Figure 1E*). The average symplectic length was significantly shorter in mutants compared to wild types. The among-individual variation was significantly greater in mutants compared to wild types. When we examined within-individual variation by determining the absolute value of the difference between left- and right-symplectic cartilage length, we found that mutants have more within-individual variation than the wild type (*Figure 1F*).

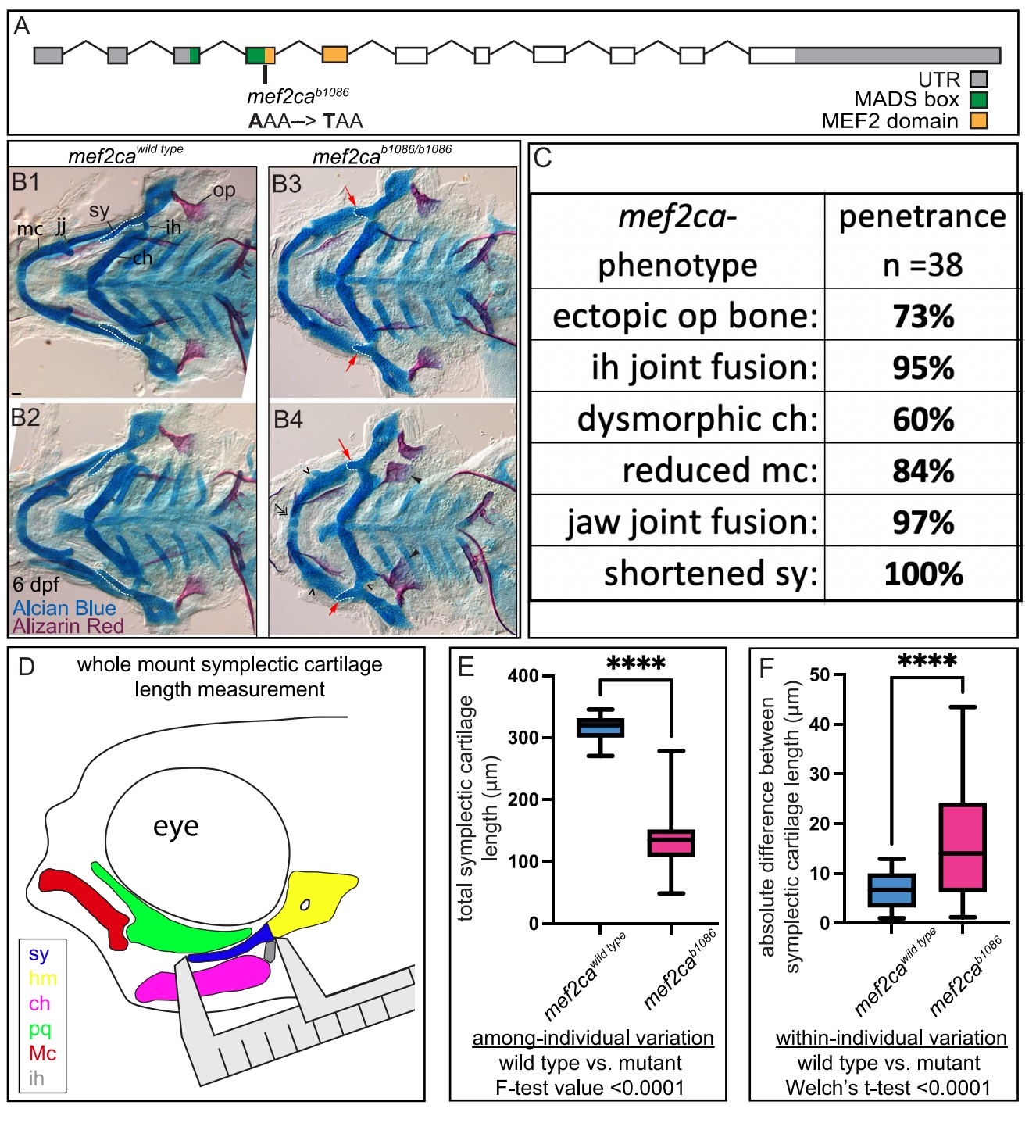

**Figure 1.** *mef2ca* mutant craniofacial phenotypes are more variable than wild types. (**A**) Schematic of *mef2ca* exonic structure. The *mef2ca*^b1086 mutant allele used in this study and regions encoding known functional domains are annotated. (**B1–B4**) Zebrafish heterozygous for *mef2ca* were pairwise intercrossed, and 6 days post fertilization (dpf), larvae were stained with Alcian blue and Alizarin red to label cartilage and bone. The individuals were then genotyped, flat mounted, and imaged. Two examples (upper and lower) are provided for each genotype. The following craniofacial skeletal elements are indicated in a wild-type individual: opercle bone (op), branchiostegal ray (br), Meckel's (mc), ceratohyal (ch), symplectic (sy) cartilages, interhyal (ih), and jaw (jj) joints. Indicated phenotypes associated with *mef2ca* mutants include: ectopic bone (arrowheads), interhyal and jaw-joint fusions (^), reduced mc (double arrowhead), and a shortened sy (red arrows). Dashed outline indicates symplectic cartilage. Scale bar: 50 µm. (**C**) The penetrance of *mef2ca* mutant-associated phenotypes observed in 6 dpf homozygous mutant larvae is indicated. (**D**) Schematic indicating how the

*Figure 1 continued on next page*

Figure 1 continued

symplectic cartilage length was measured in this study. (**E**) Symplectic cartilage length was measured from 6 dpf wild type or homozygous mutant larvae. The p-value from a Welch's t-test is indicated (****≤0.0001). F-test value testing for significant differences in variation between genotypes is indicated. (**F**) Symplectic cartilage length on left and right sides of 6 dpf zebrafish was measured to determine fluctuating asymmetry, or the absolute difference between left and right, for wild type or mutant larvae. The p-value from a Welch's t-test is indicated (****≤0.0001). For box and whisker plots, the box extends from the 25th to 75th percentiles. The line in the middle of the box is plotted at the median, and the bars are minimum and maximum values. For E and F, n=22 for wild types and 44 for mutants.

The online version of this article includes the following source data and figure supplement(s) for figure 1:

**Source data 1.** Penetrance data and raw symplectic cartilage length measurements.

**Figure supplement 1.** Symplectic cartilage length is a good proxy for other craniofacial phenotypes.

## Selective breeding reveals that severity and among-individual variation, but not within-individual variation, is heritable and segregates with penetrance

Another strength of the zebrafish system is that penetrance can be altered by selective breeding to better understand the influence of genetic background on severity and variation. An ongoing, long-term selective-breeding experiment in our laboratory derived two strains of zebrafish with consistently low and high penetrance of *mef2ca*-associated phenotypes in *mef2ca*^b1086^ homozygous mutants (*Nichols et al., 2016*; *Brooks and Nichols, 2017*; *Figure 2A*). Our previous work examined the phenotypes present in homozygous mutant fish from these strains (*Sucharov et al., 2019*). Here, we closely examined *mef2ca* wild types (*mef2ca*^+/+^) from these strains. We were surprised to observe that occasionally some *mef2ca*-associated phenotypes, like shortened symplectic cartilages (19%, n=16) and fused interhyal joints (13%, n=16), were observed in *mef2ca*^+/+^ individuals from the high-penetrance strain (*Figure 2B*). In contrast, we never observed these phenotypes in *mef2ca*^+/+^ individuals from the low-penetrance strain. Importantly, these phenotypes are not likely due to general developmental delay because other skeletal structures like pharyngeal teeth are unaffected. Thus, the phenotypes we discovered in high-penetrance *mef2ca*^+/+^ individuals are specific to developmental processes associated with *mef2ca* function. Quantifying the severity of the shortened symplectic cartilage phenotype, we found significant differences between low- and high-penetrance strains for both *mef2ca*^+/+^ as well as *mef2ca*^+/-^, but homozygous mutants were statistically the same by this measure in the two strains (*Figure 2C*).

When we examined variation in symplectic cartilage length, we compared like genotypes between strains and observed that among-individual variation was greater in the high-penetrance strain for homozygous wild types and heterozygotes, and there was no significant difference in variation between strains for homozygous mutants (*Figure 2D and E*). Thus, the genetic background producing more severe phenotypes also exhibited more among-individual variation, even in wild types. Within-individual variation was not significantly different between backgrounds (*Figure 2F and G*).

## Six *mef2* paralogs in the zebrafish genome share highly conserved amino acid sequences

We sought to explore potential mechanisms underlying the differences in severity and variation between strains. Work from budding yeast demonstrates that gene duplication contributes to genetic robustness against null mutations and that the probability of compensation by gene duplicates is correlated with sequence similarity among the gene duplicates (*Gu et al., 2003*). Therefore, we hypothesized that *mef2ca* duplicates with high sequence similarity would modulate severity and variation. There are five *mef2ca* paralogs in the zebrafish genome. To determine which are the most similar to *mef2ca* and therefore more likely to compensate for its loss, we used Clustal Omega to compare the amino acid sequences of *mef2* paralogs. While the C-terminal domain is divergent among different *mef2* genes (*Molkentin et al., 1996*; *Figure 3—figure supplement 1*), the zebrafish *mef2* paralogs each encode a MADS box and MEF2 domain which are remarkably similar in sequence across all paralogs. We found that the paralog with the most sequence similarity to *mef2ca* is *mef2cb* (*Figure 3A and B*). Together, these genes are the co-orthologs of mammalian *Mef2c*. *mef2aa* and *mef2ab*, co-orthologs of mammalian *Mef2a*, also share similar sequences to each other and the *mef2c* pair. By these analyses, *mef2d* and *mef2b* sequences are more divergent. There is only one zebrafish ortholog for *Mef2d* and *Mef2b*,

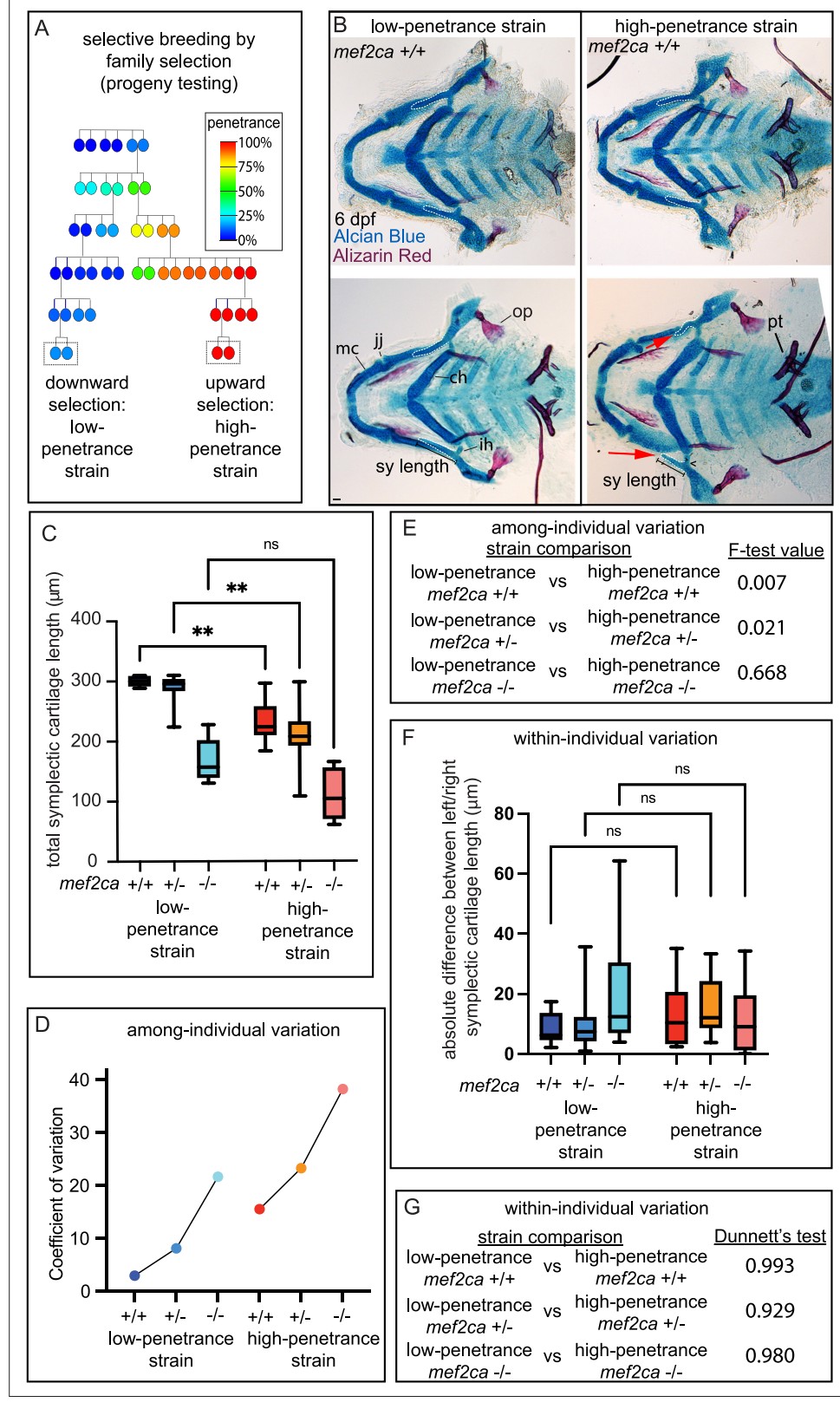

**Figure 2.** Selective breeding affects *mef2ca*-associated phenotype severity and variation in *mef2ca* wild types, heterozygotes, and homozygous mutants. (**A**) Selective breeding pedigree illustrating ectopic bone phenotype penetrance inheritance. Dashed boxes indicate families used in this study. Six generations of full-sibling inbreeding produced the animals used here, a more complete pedigree extending back >10 generations can be found in our

*Figure 2 continued on next page*

previous publication (**Sucharov et al., 2019**). (**B**) Alcian blue- and Alizarin red-stained animals from the low- and high-penetrance strains were genotyped, and *mef2ca* homozygous wild types were flat mounted and imaged. The following craniofacial skeletal elements are indicated in a wild-type individual from the low-penetrance strain: opercle bone (op), Meckel's (mc), ceratohyal (ch), symplectic (sy) cartilages, interhyal (ih), and jaw (jj) joints. Phenotypes normally associated with *mef2ca* homozygous mutants are present in some wild types from the low-penetrance strain including: ih joint fusions (^) and shortened sy (red arrows). Bars indicating sy length are presented to illustrate the shortened symplectic phenotype present in some high-penetrance wild types but not low-penetrance wild types. A stage-appropriate complement of ankylosed of pharyngeal teeth (pt) are present, and normal sized op bones are present in the individual with shortened sy, indicating the phenotypes we discovered in high-penetrance *mef2ca*[+/+] are not due to general delay. Dashed outline indicates symplectic cartilage. Scale bar: 50 μm (**C**) Symplectic cartilage length was measured from 6 days post fertilization (dpf) larvae from wild types, heterozygotes, and homozygous mutants from both the low- and high-penetrance strains. p-Values from a Dunnet's T3 test are indicated (**≤0.01). (**D**) The coefficient of variation for symplectic length in all three genotypes from both strains was plotted (**E**) Table listing F-test values testing for significant differences in variation between strains comparing the same genotype. (**F**) Symplectic cartilage length on left and right sides of 6 dpf zebrafish was measured to determine fluctuating asymmetry or the absolute difference between left and right for all three genotypes from both strains. (**G**) Table listing the Dunnett's test for significant differences in fluctuating asymmetry between all three genotypes from both strains. For box and whisker plots, the box extends from the 25th to 75th percentiles. The line in the middle of the box is plotted at the median, and the bars are minimum and maximum values.

The online version of this article includes the following source data for figure 2:

**Source data 1.** Symplectic cartilage length measurements from selectively bred strains.

presumably the other co-ortholog for each was lost. These results and interpretations are consistent with previous analyses of the evolutionary history of the *mef2* family (**Chen et al., 2017**; **Wu et al., 2011a**; **Martindale et al., 2004**; **He et al., 2019**; **Wu et al., 2011b**). Thus, these duplicates are strong candidates for mechanistically underlying the differences we observe between strains.

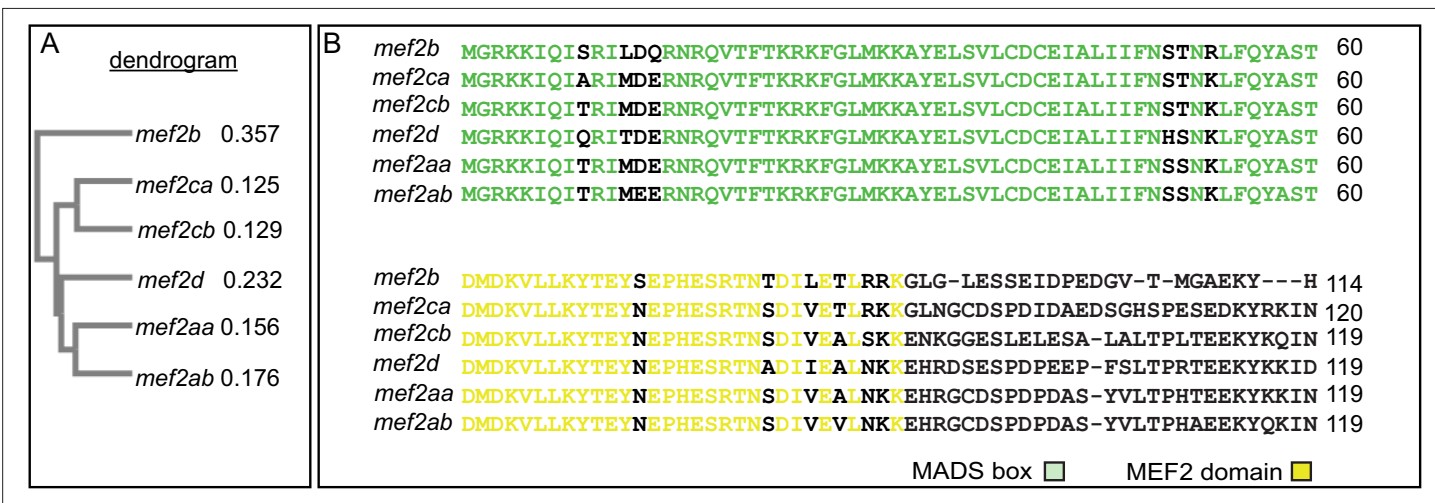

**Figure 3.** Zebrafish *mef2* paralogs encode highly conserved MADS box (MCM1, agamous, deficiens, and SRF) and MEF2 domains. (**A**) Neighbor joining tree generated by Clustal Omega Multiple Sequence Alignment tool depicts the evolutionary relationships between the different zebrafish *mef2* paralogs. The distance values (branch length) are indicated, which represent the evolutionary distance between the individual amino acid sequences and a consensus sequence. (**B**) *mef2*-encoded protein sequence alignment reveals high conservation of MADS box (green) and MEF2 (yellow) domains among all six paralogs. These domains are responsible for DNA binding, dimerization, and cofactor interactions. Transcript IDs used for alignment using the HHalign algorithm are listed in the Materials and methods.

The online version of this article includes the following figure supplement(s) for figure 3:

**Figure supplement 1.** Zebrafish *mef2* paralogs encode highly conserved N-terminal MADS box (MCM1, agamous, deficiens, and SRF) and MEF2 domains but divergent C-terminal domains.

## Closely related *mef2* paralogs share similar temporospatial expression dynamics

Our previous work, and that of others, demonstrates when (20–30 hpf) (hours post fertilization) and where (cranial neural crest cells) *mef2ca* functions during craniofacial development (*De Laurier et al., 2014*; *Sucharov et al., 2019*; *Miller et al., 2007*). We therefore hypothesized that *mef2ca* duplicates with temporospatial expression dynamics similar to *mef2ca* would modulate severity and variation. To examine paralog temporal expression dynamics, we performed RT-quantitative PCR (qPCR) (quantitative polymerase chain reaction) on wild-type zebrafish heads from 20 to 30 hpf (*Figure 4A*). We found that closely related paralogs share similar gene expression dynamics. For example, *mef2ca* and *mef2cb* have both early (~24 hpf) and late (~28 hpf) expression peaks. *mef2aa* and *mef2ab* share an early expression peak. *mef2d* and *mef2b* have a single late and early expression peak, respectively.

To examine paralog expression in the cells that give rise to the craniofacial skeleton, we examined our previously published single cell RNA-sequencing dataset to monitor *mef2* paralog expression in isolated 24 hpf wild-type cranial neural crest cells (*Mitchell et al., 2021*). While this method does not capture all the cells orchestrating craniofacial development, it does allow us to specifically assay cranial neural crest cells, including those residing in the anterior arches, which are the precursors of the cells which form the skeletal elements affected by *mef2ca* loss. We found that *mef2ca* has the strongest expression of all the paralogs across different populations of cranial neural crest cells and that expression is strongest in the anterior arches (*Figure 4B*). The closely related paralog *mef2cb* is the next highest expressed, while other paralogs are more weakly expressed in these craniofacial progenitor cells at this stage.

## *mef2* paralogs are differentially expressed in the low- versus high-penetrance strains, and paralog expression varies in an unselected background

Data showing that paralog transcriptional adaptation does not account for the phenotypic differences between the low- and high-penetrance strains (*Sucharov et al., 2019*) do not rule out the possibility that selective breeding changed paralog expression between strains in *mef2ca* wild types. To examine this possibility, we used RT-qPCR to compare *mef2* paralog expression between wild types from the high- and low-penetrance strains. Strikingly, we found that *mef2aa, mef2ab, mef2b, mef2ca,* and *mef2d* were all significantly differentially expressed in heads from wild types from the high-penetrance strain compared with heads from wild types from the low-penetrance strain (*Figure 4C*). These findings strongly suggest that one outcome of selective breeding for low- and high-*mef2ca* phenotype penetrance is generally increased and decreased expression, respectively, of the *mef2* paralogs. We do not see overall increases in transcription in the low-penetrance strain compared with the high-penetrance strain; housekeeping genes are not significantly upregulated in the low-penetrance strain (*Figure 4—figure supplement 1*).

The differences in paralog expression between selectively bred strains suggest that we might be selecting upon standing variation in paralog expression present in the unselected strain. To test this, we examined paralog gene expression in heads from offspring of two separate families of unselected AB wild types. We found significant differences in *mef2ab, mef2b, mef2cb,* and *mef2d* expression between these families. Of note, the direction of paralog expression differences between families was consistent across paralogs. That is, family 1 had generally higher *mef2* paralog expression compared with family 2. These data support that paralog expression variation is present in unselected lines and that our selective breeding for penetrance enriched for these expression differences.

## *mef2cb* buffers against *mef2ca* loss

We hypothesized that the paralog expression differences that we discovered between strains underlie the differences in severity and variation between the low- and high-penetrance strains. To test this hypothesis, we studied *mef2ca* mutants in the context of *mef2* paralog mutations. For *mef2cb*, we crossed a previously generated allele (*De Laurier et al., 2014*) into the *mef2ca* low-penetrance strain. We then maintained this stock by outcrossing to unselected AB. *mef2cb* is the paralog with the highest degree of sequence similarity to *mef2ca* and is the second highest expressed *mef2* paralog in cranial neural crest cells behind *mef2ca* (*Figures 3 and 4*). We confirmed that *mef2cb* homozygous mutants do not have an overt craniofacial phenotype and are homozygous viable (*De Laurier et al., 2014*;

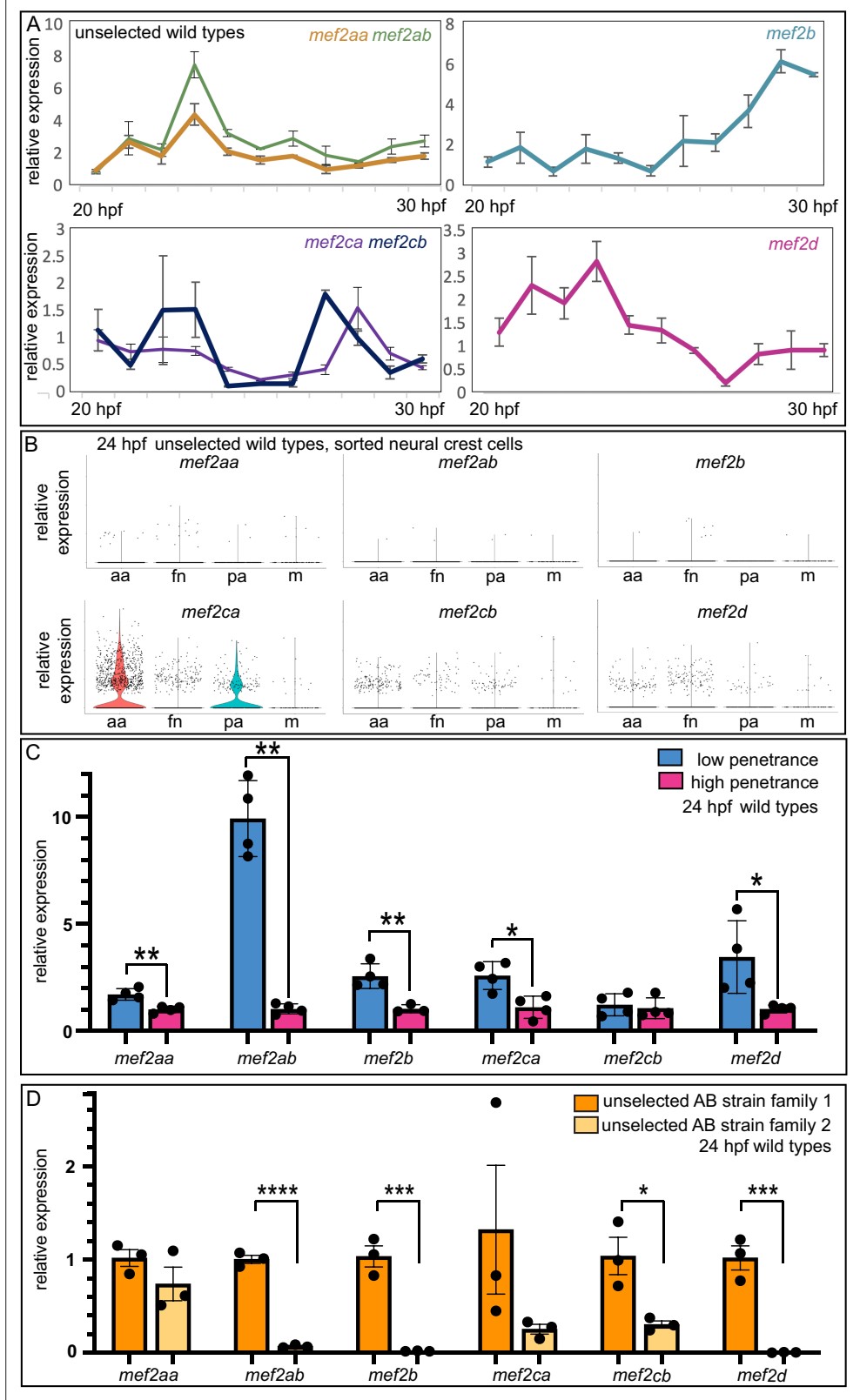

**Figure 4.** Wild-type gene expression studies reveal *mef2* paralog expression dynamics, strain-specific expression levels, and standing paralog expression variation. (**A**) Wild-type head expression of all *mef2* paralogs was quantified by RT-quantitative PCR (qPCR) at 1 hr intervals from 20 to 30 hpf. Expression of each paralog was normalized to *rps18*. Error bars are SD. (**B**) Expression of each paralog in cranial neural crest cells at 24 hpf was

*Figure 4 continued on next page*

*Figure 4 continued*

determined by single cell RNA-sequencing on sorted cells. Seurat-based clustering subdivided the cells into four populations as described (***Mitchell et al., 2021***): anterior arches (aa), frontonasal (fn), posterior arches (pa), and a satellite population containing melanocyte lineage cells (m). (**C**) Wild-type head expression of all *mef2* paralogs was quantified by RT-qPCR at 24 hpf to compare paralog expression levels between the low- and high-penetrance strains. Expression of each paralog was normalized to *rps18*. Asterisks indicate significant difference (*≤0.05 and **≤0.01). Error bars are SD. (**D**) Wild-type head expression of all *mef2* paralogs was quantified by RT-qPCR at 24 hpf to compare paralog expression levels between two families from the unselected AB strain. Expression of each paralog was normalized to *rps18*. Asterisks indicate significant difference (*≤0.05, ***≤0.001, and ****≤0.0001). Error bars are SEM.

The online version of this article includes the following source data and figure supplement(s) for figure 4:

**Source data 1.** qPCR raw values.

**Source data 2.** qPCR raw values.

**Source data 3.** qPCR raw values.

**Source data 4.** qPCR raw values.

**Figure supplement 1.** Overall transcription is not increased in the low-penetrance strain.

---

*Figure 5A, B and C*), indicating that *mef2cb* is not required for zebrafish craniofacial development in an otherwise wild-type background. However, when we removed one functional copy of *mef2ca* from *mef2cb* homozygous mutants, *mef2ca* mutant-associated phenotypes developed (***Figure 5B and C***), phenocopying the high-penetrance strain (***Sucharov et al., 2019***). We find further evidence for phenocopy; *mef2ca* homozygous mutant phenotype penetrance increased when we removed one copy of *mef2cb* (***Figure 5B and D***). Removing both copies of *mef2cb* from *mef2ca* homozygous mutants produces severe, nonspecific defects which make larvae impossible to meaningfully study (***De Laurier et al., 2014***) although their craniofacial skeletons are severely affected (***Figure 5—figure supplement 1***). Measuring the symplectic cartilage length further demonstrates that when *mef2cb* is fully functional, development is buffered against partial loss of *mef2ca*; there is no difference in symplectic length between wild types and *mef2ca* heterozygotes (wild type versus *mef2ca*$^{+/-}$;*mef2cb*$^{+/+}$). In contrast, when *mef2cb* is disabled, development is sensitive to partial loss of *mef2ca* (wild type versus *mef2ca*$^{+/-}$;*mef2cb*$^{-/-}$) (***Figure 5E***). Removing copies of *mef2cb* from *mef2ca* wild types does not significantly change symplectic cartilage length but does significantly increase symplectic cartilage variation (***Figure 5F***). Thus, even in the *mef2ca* wild-type context, this paralog buffers against phenotypic variation. We conclude that *mef2cb* buffers against *mef2ca*-associated phenotype severity, and among-individual variation, but not within-individual variation (***Figure 5D, E*** and ***Figure 5—figure supplement 2***). However, our gene expression study in high- and low-penetrance strains indicated that several *mef2* paralogs are differentially expressed between strains. Therefore, we examined how other paralogs might also buffer against *mef2ca* loss.

## *mef2d* buffers against *mef2ca* homozygous mutant severity but not variability

We detected *mef2d* transcripts in the anterior arch population of 24 hpf wild-type cranial neural crest cells, and *mef2d* expression is significantly lower in the high-penetrance strain compared with the low-penetrance strain (***Figure 4C and D***). To explore the function of this paralog, we generated a *mef2d* mutant allele (***Figure 6A***). For *mef2d* and all subsequent paralog functional experiments (*mef2b* and *mef2aa*), we generated new mutant alleles using (Clustered regularly interspaced short palindromic repeats) CRISPR/Cas9 mutagenesis in the low-penetrance strain. We then maintained these stocks by outcrossing to unselected AB. Homozygous *mef2d* mutants did not develop any overt skeletal phenotypes in an otherwise wild type, unselected background (***Figure 6B***). Intercrosses between heterozygotes produced homozygous mutant adults at the expected Mendelian frequency, indicating that *mef2d* is not required for craniofacial development or viability in laboratory conditions. Similarly, *Mef2d* mutant mice are viable and display no overt phenotypic abnormalities in standard laboratory conditions (***Arnold et al., 2007***; ***Kim et al., 2008***). To test our hypothesis that reduced levels of *mef2d* expression associated with selective breeding for high *mef2ca* penetrance contribute to severity, we examined offspring from *mef2ca*;*mef2d* double heterozygous parents (***Figure 6B***). When

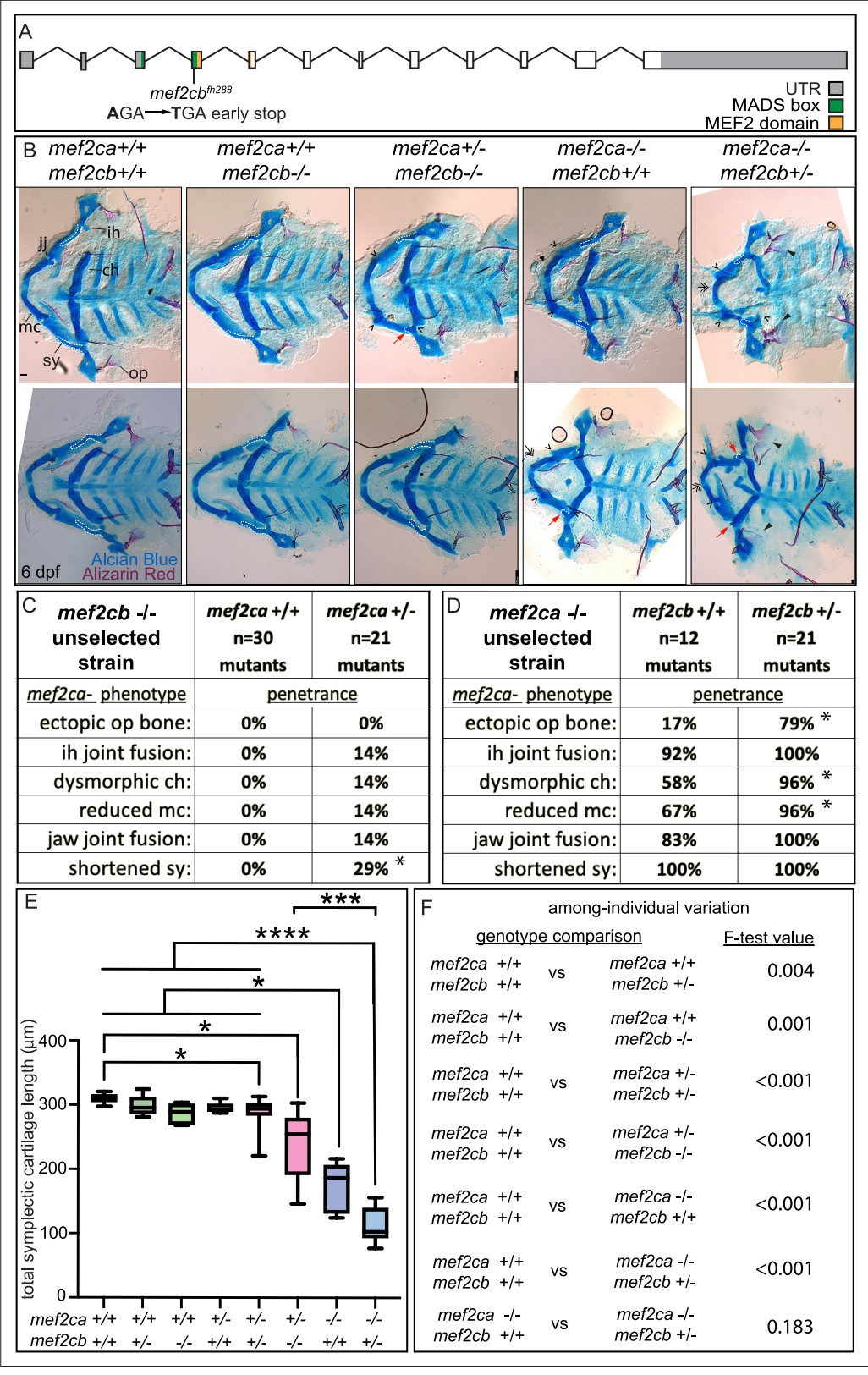

**Figure 5.** *mef2cb* function buffers against *mef2ca* loss. (**A**) Schematic of *mef2cb* exonic structure, mutant allele used in this study, and regions encoding proposed functional domains are annotated. (**B**) Zebrafish heterozygous for both *mef2ca*[b1086] and *mef2cb* were pairwise intercrossed. 6 days post fertilization (dpf) larvae were stained with Alcian blue and Alizarin red to label cartilage and bone. Stained larvae were genotyped, flat mounted, and

*Figure 5 continued*

imaged. The following craniofacial skeletal elements are indicated in a wild-type individual: opercle bone (op), branchiostegal ray (br), Meckel's (mc), ceratohyal (ch), symplectic (sy) cartilages, interhyal (ih), and jaw (jj) joints. Indicated phenotypes associated with *mef2ca* mutants include: ectopic bone (arrowheads), interhyal and jaw-joint fusions (^), dysmorphic ch (arrows), reduced mc (double arrowhead), and a shortened sy (red arrows). Dashed outline indicates symplectic cartilage. Scale bar: 50 μm (**C** and **D**) The penetrance of *mef2ca* mutant-associated phenotypes observed in 6 dpf larvae is indicated. Asterisk indicates significant difference in penetrance between the indicated genotypes by Fishers exact test. (**E**) Symplectic cartilage length was measured from 6 dpf larvae from the indicated genotypes. Asterisks indicate significant differences in symplectic length. The p-values from a Dunnet's T3 test are indicated (*≤0.05, ***≤0.001, and ****≤0.0001) (**F**) Table listing F-test values for significant differences in variation between genotypes. For box and whisker plots, the box extends from the 25th to 75th percentiles. The line in the middle of the box is plotted at the median, and the bars are minimum and maximum values. N's for all analyses are indicated in C and D.

The online version of this article includes the following source data and figure supplement(s) for figure 5:

**Source data 1.** Symplectic length cartilage measurements.

**Figure supplement 1.** Zebrafish *mef2ca;mef2cb* double homozygous mutants develop nonspecific developmental defects.

**Figure supplement 2.** Increased severity is not associated with increased within-individual.

we examined the penetrance of all *mef2ca* mutant-associated skeletal phenotypes in *mef2ca* homozygous mutants, we found that the frequency of ventral cartilage defects was significantly increased by removing a single functional copy of *mef2d* and further increased in double homozygous mutants (*Figure 6C*). The penetrance of other *mef2ca*-associated phenotypes was not significantly changed when *mef2d* function was removed. While symplectic length was not affected in *mef2d* single mutants, we found that removing both copies of *mef2d* from *mef2ca* homozygous mutants significantly shortened symplectic cartilage length. We did not observe significant changes in among-individual variation when we removed *mef2d* function from *mef2ca* homozygotes, but symplectic cartilage length is more variable in *mef2ca* heterozygotes when *mef2d* is mutated (*Figure 6E*).

## *mef2b* buffers against *mef2ca* homozygous mutant variation but not severity

*mef2b* is the most divergent *mef2ca* paralog by amino acid sequence and is only minimally expressed in cranial neural crest cells at 24 hpf (*Figures 3 and 4*). However, we did observe significantly higher *mef2b* expression in the low-penetrance strain compared with the high-penetrance strain (*Figure 4D*). We generated a *mef2b* mutant allele to test for a role for this gene in craniofacial development (*Figure 7A*). Homozygous *mef2b* mutants did not exhibit any overt skeletal phenotypes (*Figure 7B*) and intercrosses between animals heterozygous for this allele produced homozygous mutant adults at the expected Mendelian frequency. When we removed functional copies of *mef2b* from *mef2ca* homozygous mutants, we did neither observe any significant changes in *mef2ca* mutant-associated phenotype penetrance (*Figure 7C*) nor did we detect further reductions in symplectic cartilage length when functional copies of *mef2b* were removed from *mef2ca* homozygous mutants (*Figure 7D*). These findings indicate that *mef2b* function does not affect *mef2ca* mutant severity. However, we do observe a modest increase in among-individual variation in *mef2ca* homozygous mutants when *mef2b* is disabled (*Figure 7E*).

## *mef2aa* buffers against *mef2ca* heterozygous phenotypes but not *mef2ca* homozygous mutant phenotypes

*mef2aa* is minimally expressed in cranial neural crest cells (*Figure 4*). However, *mef2aa* expression differs between strains (*Figure 4C*). We generated a *mef2aa* mutant (*Figure 8A*) that does not develop any overt skeletal phenotypes when homozygous (*Figure 8B*). When we removed *mef2aa* from *mef2ca* homozygous mutants, we did not observe any significant changes in *mef2ca*-associated phenotype penetrance (*Figure 8C*), symplectic cartilage length (*Figure 8D*), or variation (*Figure 8E*). However, when we removed one functional copy of *mef2ca* from *mef2aa* homozygous mutants, *mef2ca*-associated phenotypes developed with low frequency (*Figure 8B*). Specifically, some *mef2ca* heterozygous animals developed nubbins (small lumps of cartilage) and shortened symplectic cartilages;

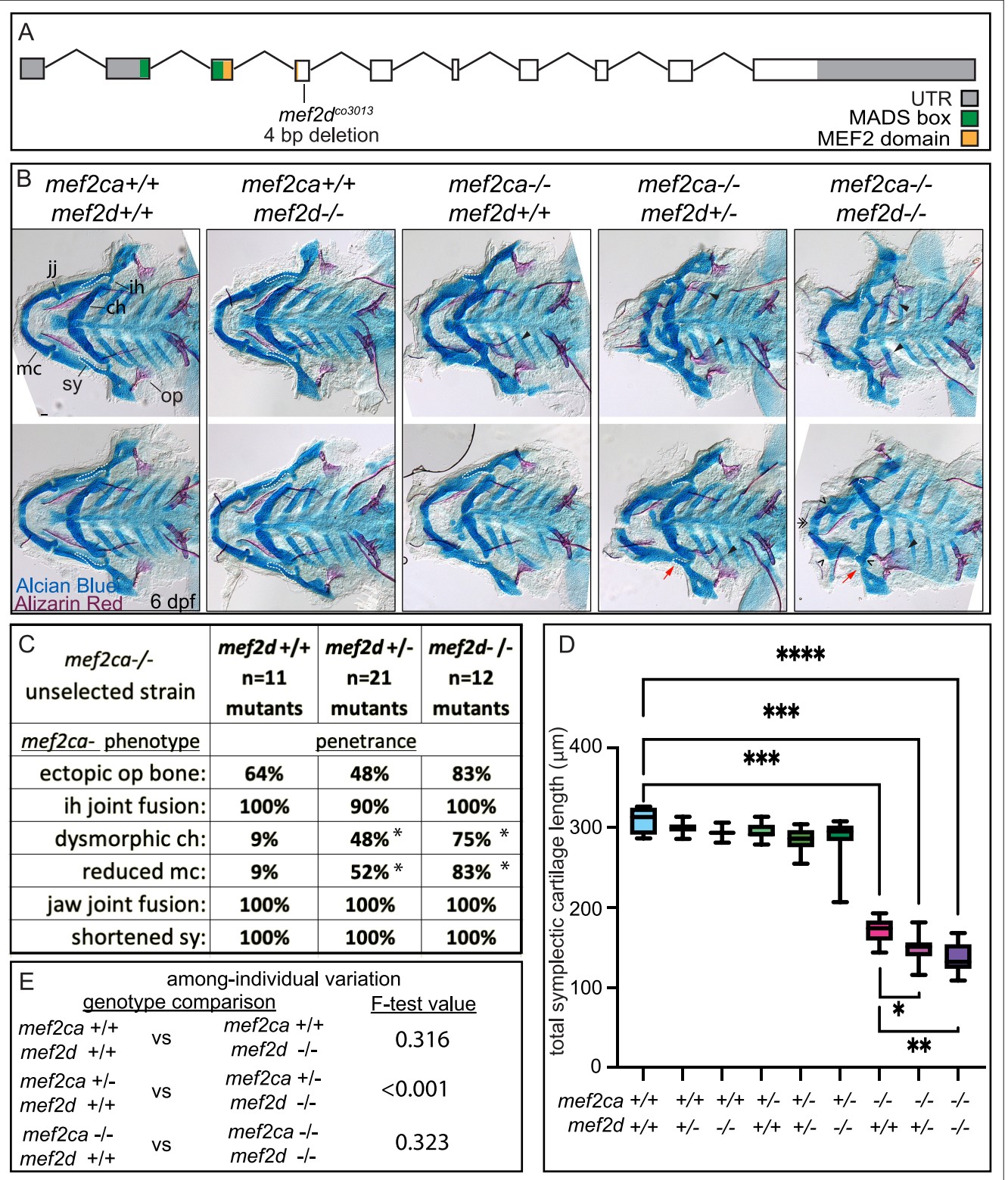

**Figure 6.** *mef2d* function buffers against *mef2ca* loss. (**A**) Schematic of *mef2d* exonic structure, mutant allele used in this study, and regions encoding proposed functional domains are annotated. (**B**) Zebrafish heterozygous for both *mef2ca* [b1086] and *mef2d* were pairwise intercrossed. 6 days post fertilization (dpf) larvae were stained with Alcian blue and Alizarin red to label cartilage and bone. Stained larvae were genotyped, flat mounted, and imaged. The following craniofacial skeletal elements are indicated in a wild-type individual: opercle bone (op), branchiostegal ray (br), Meckel's (mc), ceratohyal (ch), symplectic (sy) cartilages, interhyal (ih), and jaw (jj) joints. Indicated phenotypes associated with *mef2ca* mutants include: ectopic bone

*Figure 6 continued on next page*

*Figure 6 continued*

(arrowheads), interhyal and jaw-joint fusions (^), dysmorphic ch (arrows), reduced mc (double arrowhead), and a shortened sy (red arrows). Dashed outline indicates symplectic cartilage. Scale bar: 50 μm (**C**) The penetrance of *mef2ca* mutant-associated phenotypes observed in 6 dpf larvae is indicated. Asterisk denotes significant difference in penetrance between the indicated genotypes by Fishers exact test. (**D**) Symplectic cartilage length was measured from 6 dpf larvae from the indicated genotypes, and asterisk indicates significant differences in symplectic length (*≤0.05, **≤0.01, and ***≤0.001). (**E**) Table listing F-test values testing for significant differences in variation between genotypes. For box and whisker plots, the box extends from the 25th to 75th percentiles. The line in the middle of the box is plotted at the median, and the bars are minimum and maximum values. N's for all analyses are indicated in C.

The online version of this article includes the following source data for figure 6:

**Source data 1.** Symplectic cartilage length measurements.

phenotypes traditionally only associated with homozygous mutants (*Piotrowski et al., 1996*). Neither of these phenotypes are ever seen in unselected *mef2ca* heterozygotes but do develop in *mef2ca* heterozygotes from the high-penetrance strain (*Sucharov et al., 2019*). Therefore, disabling *mef2aa* partially phenocopies the high-penetrance strain.

## Discussion
### Vestigial paralog expression may provide developmental robustness

Vertebrate *mef2* functions downstream of endothelin signaling in the developing craniofacial skeleton (*Miller et al., 2007*; *Verzi et al., 2007*). The endothelin pathway was subfunctionalized following whole genome duplications in vertebrates (*Square et al., 2020*). Thus, *mef2* genes may have been subfunctionalized following genome duplications. Comparing mice and zebrafish further supports *mef2* subfunctionalization. In mice, *Mef2c* is required for both heart and craniofacial development (*Verzi et al., 2007*; *Lin et al., 1997*). In zebrafish, both co-orthologs (*mef2ca* and *mef2cb*) function redundantly in the heart (*Hinits et al., 2012*), while craniofacial function has been subfunctionalized to just *mef2ca* (*Figure 9*). However, it is possible that the ancestral craniofacial function and expression pattern of *mef2cb* are partially retained following subfunctionalization, even though it is no longer required for this function. We propose that while duplicated genes can evolve new expression domains and functions, vestiges of their original expression pattern remain and can buffer against loss of another paralog. Consistently, with the exception of *mef2ca*, the *mef2* paralog mutants we analyzed here do not have an overt single mutant skeletal phenotype but do modify *mef2ca* mutant phenotypes. In our system, selective breeding likely fixed existing paralog expression variation. Thus, in the low-penetrance strain, selection may replicate paralog ancestral expression, restoring near full redundancy (*Figure 9*).

Our experiments support a model where we selected upon pre-existing alleles that either amplify or dampen existing vestigial paralog expression variation, depending on the direction of selection. We found that expression of several paralogs was affected by selection. Previously however, we reported that the rapid response to selection in our system indicates that relatively few heritable factors contribute to penetrance (*Nichols et al., 2016*; *Sucharov et al., 2019*). To account for this paradox, we propose that a few shared genetic factors regulate paralog expression. By selecting for penetrance, expression of several paralogs was increased or decreased via selection for these shared genetic factors. In support, we found evidence of shared gene regulation in our experiments examining *mef2* gene expression dynamics. The expression dynamics of paralogs closely related in sequence and phylogeny are similar, suggesting shared regulatory modules. In this model, few factors are inherited that control the expression of many paralogs. These *mef2* paralog regulatory factors might be either shared *mef2* enhancers (in cis) or might be in shared upstream transcriptional regulators (in trans) that simultaneously regulate several *mef2* paralogs. Future studies will discern between these two, not mutually exclusive, possibilities.

### Evidence for genetic assimilation

It is fascinating that a phenotype originally associated with a homozygous loss of function mutation can appear in homozygous wild types after selective breeding for high penetrance. We previously observed a similar phenomenon with heterozygotes in the high-penetrance strain and proposed that this is a form of genetic assimilation (*Sucharov et al., 2019*; *Waddington, 1953*). In Waddington's

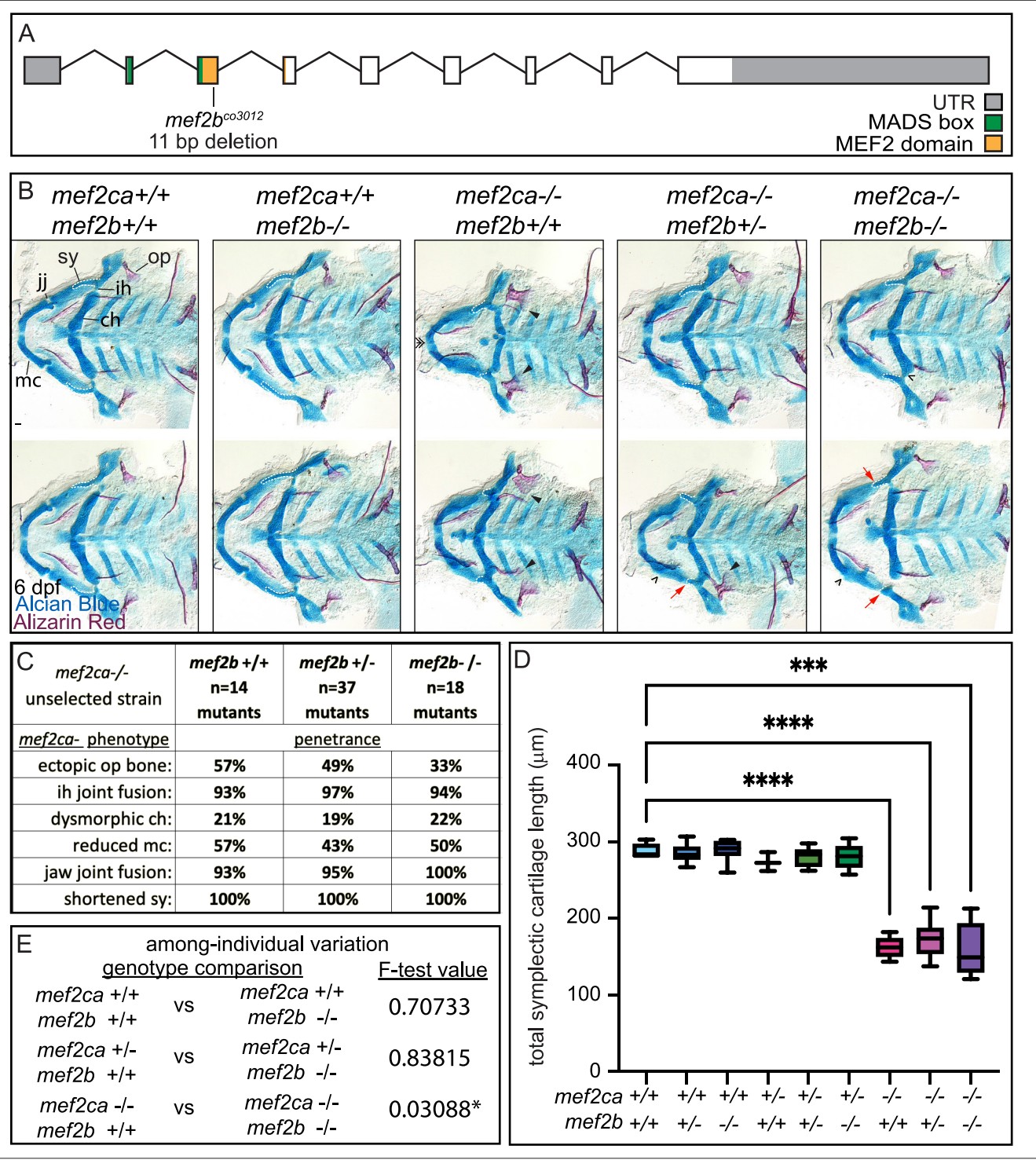

**Figure 7.** *mef2b* function buffers against *mef2ca* loss. (**A**) Schematic of *mef2b* exonic structure, mutant allele used in this study, and regions encoding proposed functional domains are annotated. (**B**) Zebrafish heterozygous for both *mef2ca* [b1086] and *mef2b* were pairwise intercrossed. 6 days post fertilization (dpf) larvae were stained with Alcian blue and Alizarin red to label cartilage and bone. Stained larvae were genotyped, flat mounted, and imaged. The following craniofacial skeletal elements are indicated in a wild-type individual: opercle bone (op), branchiostegal ray (br), Meckel's (mc), ceratohyal (ch), symplectic (sy) cartilages, interhyal (ih), and jaw (jj) joints. Indicated phenotypes associated with *mef2ca* mutants include: ectopic bone (arrowheads), interhyal and jaw-joint fusions (^), dysmorphic ch (arrows), reduced mc (double arrowhead), and a shortened sy (red arrows). Scale bar: 50 μm. (**C**) The penetrance of *mef2ca* mutant-associated phenotypes observed in 6 dpf larvae is indicated. (**D**) Symplectic cartilage length was measured from 6 dpf larvae from the indicated genotypes, and asterisk indicates significant differences in symplectic length (***≤0.001 and ****≤0.0001). (**E**) Table

*Figure 7 continued on next page*

*Figure 7 continued*

listing F-test values testing for significant differences in variation between genotypes. For box and whisker plots, the box extends from the 25th to 75th percentiles. The line in the middle of the box is plotted at the median, and the bars are minimum and maximum values. N's for all analyses are indicated in C.

The online version of this article includes the following source data for figure 7:

**Source data 1.** Symplectic cartilage length measurements.

genetic assimilation experiments (*Waddington, 1942*; *Waddington, 1953*; *Waddington, 1959*; *Waddington, 1952*; *Waddington, 1956*; *Waddington, 1957*), he studied phenotypes that were originally only present in perturbed conditions. Following selective breeding, the phenotypes arose without the perturbation. Similarly, we studied phenotypes that were originally only present in perturbed conditions, *mef2ca* mutants. Following selective breeding, the phenotypes arose without the perturbation, in wild types. We offer a mechanistic explanation for our findings. Analyzing craniofacial skeletons in the context of the *mef2ca* mutation revealed cryptic paralog expression variation. Specifically, in wild types, paralog expression variation does not impact phenotype. However, in mutants, the consequences of variable expression levels are unveiled, allowing us to select upon them. Because the paralogs seem to have shared regulatory systems, we enriched for variants producing lower expression of several paralogs, including *mef2ca*, by selective breeding for high penetrance. This pan-paralog expression decrease led the high-penetrance strain to be both extremely sensitive to coding mutations in *mef2ca* and to manifest *mef2ca* mutant-associated phenotypes in *mef2ca* wild types from this strain. The latter is likely due to decreased *mef2ca* expression in high-penetrance *mef2ca* wild types.

## Decoupling buffering mechanisms

We found that paralogs modularly buffer the *mef2ca* homozygous mutants. For example, disabling *mef2cb* affects most *mef2ca*-related craniofacial phenotypes, while *mef2d* mutations primarily affect penetrance of ventral cartilage phenotypes, and only variation is affected when *mef2b* function is removed. Although we were unable to mutagenize *mef2ab*, despite significant effort, this is not likely to affect our general conclusions. Similar to work in other systems (*Gu et al., 2003*), the paralog with the highest expression (*mef2ca*) is the one associated with a single mutant phenotype, and the paralog with the most sequence similarity (*mef2cb*) is the one playing the largest buffering role. However, others have reported that partial gene expression overlap is also predictive of buffering capacity (*Kafri et al., 2005*; *Kafri et al., 2006*), in line with our results that even distantly related paralogs with more dissimilar expression profiles (*mef2d*) can modify penetrance in our system. Different types of compensation events have previously been classified as either active or passive (*Diss et al., 2014*). In our system, compensation is likely passive, but subject to selection, which can change robustness across generations.

Determining whether the specific portion of the phenotype that each paralog buffers is related to that paralog's wild-type expression pattern would expand our understanding of modularity. Unfortunately, in situ gene expression protocols were not sensitive enough to detect the low expression of the individual paralogs. Moreover, the quantitative phenotyping in this study was limited to the symplectic cartilage, as a proxy for the whole system (*Figure 1—figure supplement 1*). However, quantitative phenotyping of other parts of the developing craniofacial complex (*Sasaki et al., 2013*; *Kimmel et al., 2015*) might reveal more information about modular buffering of severity and variation by paralogs. For example, penetrance of ceratohyal cartilage defects was affected by *mef2d*, and therefore quantitative variation of this structure, which would be difficult to measure, might also be affected by loss of *mef2d*.

Our experiments also decouple the mechanisms that buffer among- and within-individual variation. Examining selectively bred strains and the different paralog mutants demonstrates that buffering in our system only regulates among-individual variation, not within-individual variation (*Figure 5—figure supplement 2*). These two types of variation are buffered by different mechanisms in our system. Our results shed some light on the long-standing debate surrounding similarities and differences between mechanisms buffering these two types of variation (*Hallgrimsson et al., 2019*).

These studies advance our understanding of craniofacial variation. We propose that cryptic variation in vestigial paralog expression is the noise underlying variable craniofacial development. Whole genome duplications producing paralogs are not zebrafish specific or specific to this gene family. In

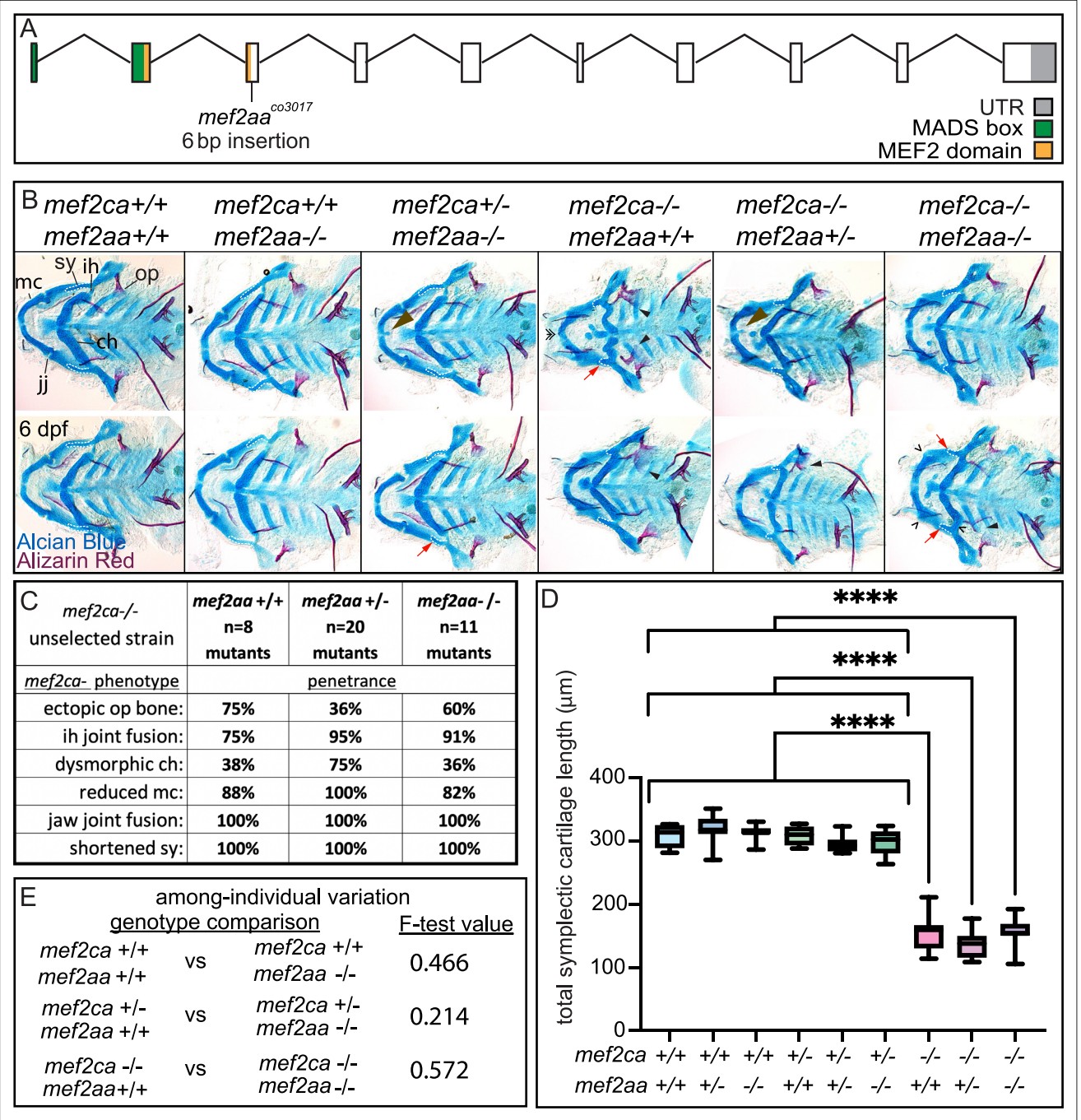

**Figure 8.** *mef2aa* function buffers against *mef2ca* partial loss. (**A**) Schematic of *mef2aa* exonic structure, mutant allele used in this study, and regions encoding proposed functional domains are annotated. (**B**) Zebrafish heterozygous for both *mef2ca* [b1086] and *mef2aa* were pairwise intercrossed. 6 days post fertilization (dpf) larvae were stained with Alcian blue and Alizarin red to label cartilage and bone. Stained larvae were genotyped, flat mounted, and imaged. The following craniofacial skeletal elements are indicated in a wild type: opercle bone (op), branchiostegal ray (br), Meckel's (mc), ceratohyal (ch), symplectic (sy) cartilages, interhyal (ih), and jaw (jj) joints. Indicated phenotypes associated with *mef2ca* mutants include: cartilage nubbin fused to the mc symphysis (brown arrowhead), ectopic bone (black arrowheads), ih and jj fusions (^), dysmorphic ch (arrows), reduced mc (double arrowhead), and a shortened sy (red arrows). Scale bar: 50 μm. (**C**) The penetrance of *mef2ca* mutant-associated phenotypes observed in 6 dpf larvae is indicated. (**D**) Symplectic cartilage length was measured from 6 dpf larvae from the indicated genotypes, and asterisk indicates significant differences in symplectic length (****≤0.0001). (**E**) Table listing F-test values testing for significant differences in variation between genotypes. For box and whisker plots, the box extends from the 25th to 75th percentiles. The line in the middle of the box is plotted at the median, and the bars are minimum and maximum values. N's for all analyses are indicated in C.

The online version of this article includes the following source data for figure 8:

**Source data 1.** Symplectic cartilage length measurements.

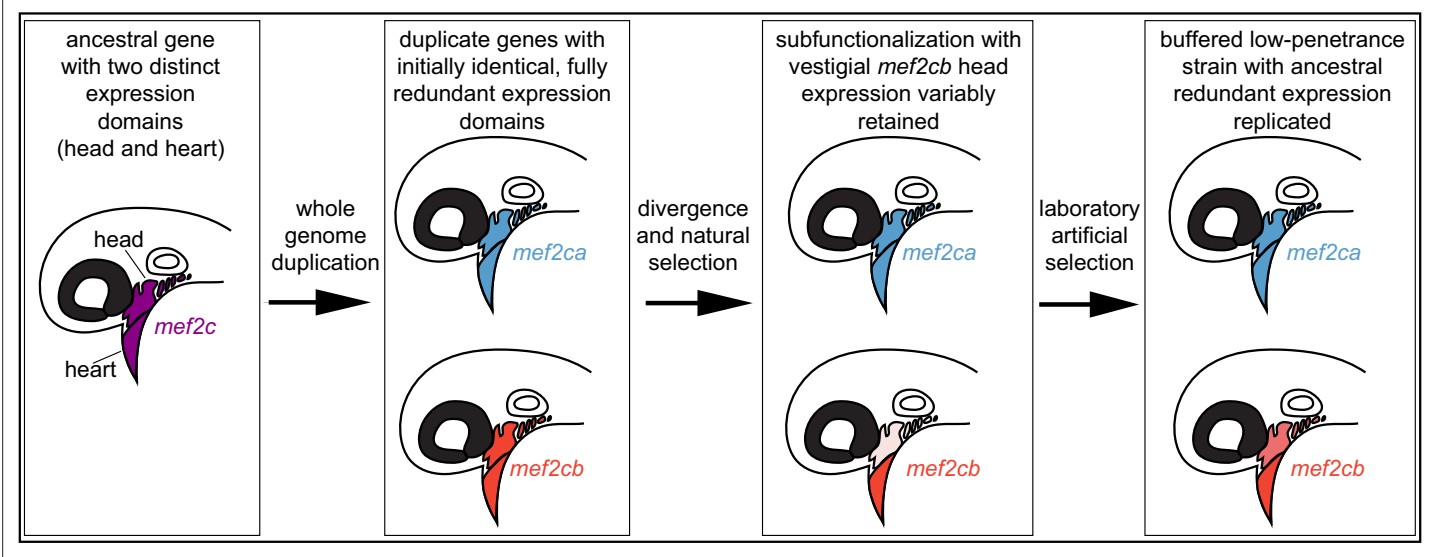

**Figure 9.** Model for *mef2* gene duplicate evolution and replication of ancestral, redundant expression via selective breeding. In our model, an ancestral *mef2* gene existed with distinct expression domains. In this example, ancestral *mef2c* had expression in the head and the heart. Following whole genome duplication, initially the regulatory and coding sequences would be identical. Through divergence and natural selection, factors controlling gene regulation would acquire mutations that dampen, but do not eliminate, some expression domains. In this example, *mef2cb* expression is dampened in the head. These gene expression changes result in subfunctionalization but with vestigial retention of the original expression. *mef2cb* is no longer required for craniofacial development, yet traces of the original craniofacial expression remain. The amount of vestigial expression is variable among individuals, resulting in variable buffering against *mef2ca* mutant phenotypes across a population. Our selective breeding selected on this variable expression resulting in higher paralog expression in the low-penetrance strain compared with the high-penetrance strain. Thus, *mef2* paralog expression in the low-penetrance strain resembles the ancestral condition following whole genome duplication, when both copies' expression profiles were highly similar, and the genes were redundant for craniofacial development.

fact, previous work demonstrates that paralogs contribute to robustness in diverse systems (*Gu et al., 2003*; *Kamath et al., 2003*; *De Kegel and Ryan, 2019*). Thus, even human craniofacial variation might be due to cryptic paralog expression variation and may explain how some genetically resilient humans (*Chen et al., 2016*) can overcome a deleterious mutant allele.

## Materials and methods
### Zebrafish strains and husbandry
All fish were maintained and staged according to established protocols (*Kimmel et al., 1995*; *Westerfield, 1993*). Selective breeding was performed as previously described (*Nichols et al., 2016*; *Sucharov et al., 2019*; *Brooks and Nichols, 2017*). Briefly, because the *mef2ca* mutation is lethal when homozygous, heterozygous full-sibling parents were intercrossed, and their homozygous mutant offspring were scored for penetrance when determining what family was to be propagated for the next generation. The *mef2ca*[b1086] and *mef2cb*[fh288] mutant alleles have been previously described (*Hinits et al., 2012*; *Miller et al., 2007*) and were maintained by outcrossing to the unselected AB background. When we first introduced the *mef2cb*[fh288] allele to *mef2ca*[b1086], we crossed it to the low-penetrance *mef2ca*[b1086] selectively bred strain and subsequently maintained these double heterozygotes by outcrossing to the unselected AB background.

### CRISPR/Cas9-induced mutant alleles
We generated germline mutant alleles using CRISPR/Cas9 mutagenesis (*Hwang et al., 2013*) with modifications as described (*Mitchell et al., 2021*). Briefly, we designed sgRNAs(Single-guide RNA) within or just downstream of the MADS or MEF2 domain. The XbaI-digested pT3TS-nCas9n plasmid (Addgene plasmid #46757) was used as a template to transcribe Cas9 mRNA with the T3 mMESSAGE kit (Invitrogen). We transcribed sgRNAs (see table below) from PCR-generated templates using the MEGAscript T7 Kit (Thermo Fisher Scientific). One cell-stage embryos were injected with a mix of

200 ng/µl Cas9 mRNA and 50 ng/µl of each gene-specific sgRNA. Injected embryos were raised, and founders identified by amplifying the genomic region containing the sgRNA site and identifying banding size shifts indicating insertions and/or deletions (see genotyping assay table for primers). All new paralog mutants were originally generated in the low-penetrance strain and were subsequently maintained on the AB background for at least three generations. The following sgRNAs were used: *mef2d*[co3013]: 5'-GGACAAATACCGGAAGAGCG-3'; *mef2b*[co2012]: 5'-CACGAGAGCCGCACTAACAC-3'; *mef2aa*[co3017]: 5'-TCATGGACGACCGTTTCGGC-3'.

We generated six independent sgRNAs for *mef2ab,* and none of them mutagenized this locus. Precise sequences of mutant alleles are indicated below:

| Gene | Mutant insertion (underlined) and deletion (italicised) sequence |
| --- | --- |
| *mef2d* [co3013] | Exon3: AATACCGGAAGA*TCGA*GGAGCTGGATATCCTC |
| *mef2b* [co3012] | Exon3: AACCTCAC*GAGAGCCGCAC*TAACAC |
| *mef2aa* [co3017] | Exon5: TCATGCCCCTGGACGACCGTTTCGGCAAA |

## Cartilage and bone staining and imaging

Fixed animals were stained with Alcian blue and Alizarin red as described previously (*Brooks and Nichols, 2017*; *Walker and Kimmel, 2007*). Alcian blue- and Alizarin red-stained 6 days post fertilization (dpf) skeletons were dissected and flat mounted for Nomarski imaging on a Leica DMi8 inverted microscope equipped with a Leica DMC2900 as previously described (*Nichols et al., 2013*).

## Phenotype scoring

For penetrance scoring, 6 dpf Alcian blue- and Alizarin red-stained skeletons were genotyped then scored for the proportion of animals with a given genotype that exhibit a particular phenotype. In the interest of strong rigor and reproducibility, phenotypes were scored by three observers blinded to genotype. All three agreed with the number of animals in each phenotypic class, indicating that phenotype penetrance can be reproducibly identified by different observers. For symplectic cartilage measurements, whole mount Alcian blue- and Alizarin red-stained skeletons were imaged under a transmitted light dissecting scope. Images were captured with Zeiss ZEN software. This software was then used to measure the linear distance in microns from the posterior most point of the interhyal cartilage and the distal tip of the symplectic cartilage similar to a previous study (*Talbot et al., 2016*). For each individual, the total symplectic length was calculated by summing the measurements from the left and right side. These same measurements were also used to calculate the absolute value of the left minus the right symplectic cartilage lengths to determine developmental instability for each individual. All raw phenotype data are presented in source data tables.

## Reverse transcription-quantitative polymerase chain reaction

Gene expression studies were performed as previously described (*Sucharov et al., 2019*). For the time course study from unselected AB and for comparing low- and high-penetrance wild types, live individual 24 hpf embryos from each strain had their heads removed. Decapitated bodies were genotyped to identify homozygous wild types. Heads from five to six identified homozygous wild types were pooled, and total RNA was extracted with TRI Reagent. cDNA was prepared with Superscript III from Invitrogen. qPCR experiments utilized a real-time PCR StepOnePlus system from Applied Biosystems and SYBR green. A standard curve was generated from serially diluted (1:2:10) cDNA pools, and primers with a slope of –3.3 ±0.3 were accepted. The relative quantity of target cDNA was calculated using Applied Biosystems StepOne V.2.0 software and the comparative Ct method. After surveying the expression of many housekeeping genes at multiple stages, we determined that *rps18* expression was the most consistent across stages, genotypes, and strains. Target gene expression in all experiments was normalized to *rps18*. Reactions were performed in technical triplicate, and the results represent two to six biological replicates. The following primers were used: *rps18* FW, 5'-CTGAACAGACAGAAGGACATAA-3' and *rps18* REV 5'-AGCCTCTCCAGATCTTCTC-3', *mef2ca* FW, 5'-GTCCAGAATCCGAGGACAAATA-3' and *mef2ca* REV 5'-GAGACAGGCATGTCGTAGTTAG-3', *mef2cb* FW, 5'-AGTACGCCAGCACAGATA-3' and *mef2cb* REV 5'-AGCCATTTAGACCCTTCTTTC –3', *mef2aa* FW, 5'-CCACGAGAGCAGAACCAACTC-3' and *mef2aa* REV 5'-GTCCATGAGGGGACTGTGAC-3', *mef2ab* FW, 5'-AACCTCACGAGAGCAGAACC-3' and *mef2ab*

REV 5'-AGGACATATGAGGCGTCTGG-3', *mef2b* FW, 5'-CCGATATGGACAAAGTGCTG-3' and *mef2b* REV 5'-CCAATCCCAATCCTTTCCTT-3', *mef2d* FW, 5'-TTCCAGTATGCCAGCACTGA-3' and *mef2d* REV 5'-CGAATCACGGTGCTCTTTCT-3'. All qPCR numerical data and statistical analyses are reported in supplementary data table.

## scRNA-seq analysis

We analyzed our published data set (*Mitchell et al., 2021*) from 24 hpf wild-type AB zebrafish cranial neural crest cells for *mef2* paralog expression. Seurat's 'VlnPlot' function was used to generate *mef2* paralog plots indicating the distribution of cells expressing each paralog across each of the four clusters we described in the previous publication. The raw, feature-barcode matrix for this dataset can be accessed from the GEO database (accession number GSE163826).

## Genotyping assays

*mef2cab1086* was genotyped by KASP as previously described (*Brooks and Nichols, 2017*). PCR-based genotyping assays are as follows:

| Gene[allele] | Primers | Enzyme | wt | mut |
|---|---|---|---|---|
| *mef2d*[co3013] | Two separate reactions are run per sample.<br>First reaction: Fw Full, Rv Full, and Fw wt<br>Second reaction: Fw Full, Rv Full, and Rev mut<br>Fw Full: 5'-AAGAAAGGCTTTAACGGTTGC-3'<br>Rv Full: 5'-AAGAGAAGGACGGAGGTTAGA-3'<br>Fw wt: 5'-GACAAATACCGGAAGAGCGA-3'<br>Rv mut: 5'-AGAGGATATCCAGCTCCTCTTC-3' | None | 173 bp and 98 bp | 121 bp and 173 bp |
| *mef2aa*[co3017] | Fw: 5'-TTGACCCAACGGTTTACAGA-3'<br>Rv: 5'-CACAAAGCCAAGCAAAAACA-3' | NlaIII | 291 bp and 145 bp | Uncut 442 bp |
| *mef2cb*[fh288] | Fw: 5'-TCCCTGCTTCTCTCTAGGTGACATTTACATCG-3'<br>Rv: 5'-TCGTGTGGCTCGTTGTACTC-3' | TaqaI | 190 bp and 10 bp | Uncut 200 bp |
| *mef2b*[co3012] | Fw: 5'-CGAGATCGCTCTCATCATCTT-3'<br>Rv: 5'-GACATACTGGAGGTATACAGACCAAA-3' | AciI | 112 bp and 38 bp | Uncut 139 bp |

## Sequence alignments

We used the Clustal Omega tool by EMBL-EBI for multiple sequence alignment of the different paralog gene products. We obtained the following transcripts and protein products from ENSEMBL for alignments: mef2aa-206 ENSDART00000171594.2, mef2ab-201 ENSDART00000173414.2, mef2b-202 ENSDART00000166300.3, mef2ca-202 ENSDART00000099134.5, mef2cb-207 ENSDART00000183585.1, and mef2d-203 ENSDART00000132589.2ENSDART00000132589.2.

## Statistical analyses

Penetrance scores were compared using Fisher's exact test to determine significance. All scoring data and exact p-values are reported in source data tables. Studies with symplectic cartilage measurements are presented as box and whisker plots, and the box extends from the 25th to 75th percentiles. The line in the middle of the box is plotted at the median, and the bars are minimum and maximum values. We used a Welch's t-test or Dunnet's T3 test to compare total symplectic cartilage (left plus right sides) between genotypes. We used F-test to test for significant differences in variation between genotypes. For developmental instability, the absolute value of the difference between left and right symplectic cartilage lengths were grouped by genotype, and Welch's t-test, Brown-Forsythe, or Welch's ANOVA tests were used to determine significant differences in left-right asymmetry between genotypes. Power analyses were used to determine the number of animals to be examined for each experiment. All statistical analyses and exact p-values are reported in source data.

## Acknowledgements

We thank Charles Kimmel for careful reading of this manuscript, members of the department of craniofacial biology for insightful discussions, Austin Tillery for contributions in the early stages of this study, and the University of Colorado zebrafish care staff.

## Additional information

### Funding

| Funder | Grant reference number | Author |
|---|---|---|
| National Institute of Dental and Craniofacial Research | R01 DE029193 | James T Nichols |
| National Science Foundation | Graduate Research Fellowships Program 201569 | Raisa Bailon-Zambrano |
| National Institute of Dental and Craniofacial Research | F32 DE029995 | Jennyfer M Mitchell |
| National Science Foundation | Graduate Research Fellowships Program | Abigail Mumme-Monheit |

The funders had no role in study design, data collection and interpretation, or the decision to submit the work for publication.

### Author contributions

Raisa Bailon-Zambrano, Formal analysis, Investigation, Writing – review and editing; Juliana Sucharov, Conceptualization, Formal analysis, Investigation, Writing – review and editing; Abigail Mumme-Monheit, Investigation, Writing – review and editing; Matthew Murry, Amanda Stenzel, Anthony T Pulvino, Jennyfer M Mitchell, Investigation; Kathryn L Colborn, Methodology; James T Nichols, Conceptualization, Formal analysis, Supervision, Funding acquisition, Investigation, Visualization, Methodology, Writing – original draft, Project administration, Writing – review and editing

### Author ORCIDs

Raisa Bailon-Zambrano (ID) http://orcid.org/0000-0002-5848-3952
Abigail Mumme-Monheit (ID) http://orcid.org/0000-0003-0090-1418
Jennyfer M Mitchell (ID) http://orcid.org/0000-0003-4222-5235
James T Nichols (ID) http://orcid.org/0000-0002-7263-1704

### Ethics

All of our work with zebrafish has been approved by the University of Colorado Institutional Animal Care and Use Committee (IACUC), Protocol # 00188. Animals were euthanized by hypothermic shock followed by 1.5% sodium hypochlorite.

### Decision letter and Author response

Decision letter https://doi.org/10.7554/eLife.79247.sa1
Author response https://doi.org/10.7554/eLife.79247.sa2

## Additional files

### Supplementary files
- Transparent reporting form
- MDAR checklist

### Data availability

All raw data are provided in supplementary data table. Sequencing dataset have been deposited in GEO. The raw, feature-barcode matrix can be accessed from the GEO database (accession number GSE163826).

The following previously published dataset was used:

| Author(s) | Year | Dataset title | Dataset URL | Database and Identifier |
|---|---|---|---|---|
| Mitchell, et al | 2021 | The alx3 gene shapes the zebrafish neurocranium by regulating frontonasal neural crest cell differentiation timing | https://www.ncbi.nlm.nih.gov/geo/query/acc.cgi?acc=GSE163826 | NCBI Gene Expression Omnibus, GSE163826 |

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
