## [Editor Report]

In this elegant genetic study, Bailon-Zambrano et al. draw on classical genetic concepts to address the clinically pertinent question of how genetic variants in the same gene can yield wildly different phenotypes in different individuals. Specifically, this work makes significant contributions to our understanding of how a mutant phenotype may be modified by the expression levels of paralogues of the mutant gene.

---

## [Decision Letter]

**Decision letter after peer review:**

Thank you for submitting your article "Variable paralog expression underlies phenotype variation" for consideration by *eLife*. Your article has been reviewed by 3 peer reviewers, and the evaluation has been overseen by a Reviewing Editor and Christian Landry as the Senior Editor. The reviewers have opted to remain anonymous.

Essential revisions:

The three reviewers agree that the question addressed is of great interest to evolutionary and developmental biologists in general and to those studying the evolution of developmental mechanisms in particular. The manuscript reports substantial and interesting advances, but in its current form it is difficult to read, which led to significant frustration among the reviewers. The manuscript needs to be substantially rewritten and shortened. The writing and interpretation of the results needs to be clarified as many statements are perceived as lacking precision. One of many examples is the statement that each selected strain has similar inbredness. However, there is no evidence that this was actually tested. The authors should also ensure that their conclusions match the results and are not over interpreted, pointing out explicitly caveats where necessary. Finally, the results need to be discussed in relation to what has been published, for example how does the phenomenon studied here relate to studies of genetic modifiers and what has been learned there.

*Reviewer #1 (Recommendations for the authors):*

P3 abstract: 'In support, mutagenizing all mef2ca paralogs in the low penetrance strain'

But one wasn't mutated. It would be helpful to say which paralogs were mutated in the abstract just so that literature searches by people studying the paralogs would be more likely to find this paper.

P3 'Human craniofacial variation allows us – and even computers -- to identify each other.

Re: 'facial variation persists among genetically homogeneous populations1.

Text should specifically talk about identical twins here.

P4 paragraph should start with a topic sentence: 'Variation can be talked about in two different ways: penetrance is….'

P5 ‘large deletion encompassing the MEF2C locus’

Is that in homozygous or heterozygous condition?

P5 text says that mef2cb mutants are viable without craniofacial phenotypes, but do they have other phenotypes?

P6 'Because no other zebrafish *mef2* gene mutants'. Reader needs to know how many *mef2* genes zebrafish has and their relationship to human *MEF2* genes.

P6 'nearly all mef2ca associated phenotypes changed penetrance as well34'

Did they change in the same direction? All more severe or all less severe?

P6 'the spread in symplectic length'

Define what this 'spread' means in phenotypic terms.

P6 'linear measurements of the symplectic cartilage, allowing us to measure severity,

among-individual variation, and within-individual variation in the zebrafish craniofacial skeleton'

Authors need to demonstrate that measuring the length of this single skeletal element is sufficient to capture the variation they intend to study.

P7 'upregulated in the low penetrance strain compared with the high-penetrance strain'

With respect to what internal control?

P7 'partially due to individuals inheriting different mutant alleles retaining different levels

of functional activity'

Right, for protein function, but also for different transcriptional expression domains or levels.

P7 'The C-terminal domain is more divergent among different *mef2* genes38.'

This paragraph is talking about alleles of one gene but this sentence is talking about different genes, so it's a bit confusing and seems out of place.

P7 'destroys the initiating methionine35.'

Is any protein made from an internal methionine that might provide an alternative initiation site?

P8 'deletion allele found in a [mildly affected] patient23'

P8 'among-individual variation associated with the opercle bone phenotype; one individual

has phenotypically wild-type opercles while the other individual has bilateral mutant

phenotypes'

Give each of the individuals in Figure 1B a different panel number, like B1, B2 for the two wild types. Then in the sentence above call them out by name so the reader doesn't have to figure out which is which.

P8 'within-individual (left-right) opercle phenotype variation is present in one of the mef2cab631 animals (lower)

Be a bit more specific, saying that in the bottom individual (which could be called B4), the left opercle is relatively normal but the right opercle is partially duplicated (or something like that).

Figure 1D. I think it would be helpful for the reader to label all of the skeletal elements in Figure 1D and highlight the symplectic by color.

P8 'Ordering our allelic series by expressivity [of this one element] shows'

What about the other phenotypes? Do they follow the same allelic series? It's possible that different alleles could affect differentially different phenotypes.

P8 'expected to be [milder] than the full deletion'

P8 'the PTC (mef2cab1086) allele would be expected to be more mild than the full deletion

(mef2caco3008) allele if transcriptional adaptation was a factor.'

Yes, agreed.

P9 'the most severe allele is also the most variable (Figure 1F).'

Consider plotting a measure of severity on the vertical axis at the right of the figure on the same horizontal axis to make the point about the correlation.

P9 'there is no significant difference [among] mutant alleles for this type of variation '

P9 'Severity is positively correlated with variation when comparing'

Would it make more sense to say 'Variation is positively correlated with severity when comparing'? Both of course are correct but phrase below says it in this order and biologically it seems to make more sense to me that the severity is a property of the allele and that the variation is a property of the biology arising from the altered protein function.

P9 'we hypothesized that [a single] mutant allele, mef2cab1086,'

P9 'shortened symplectic cartilages and fused interhyal joints, were occasionally observed'

In nearly all of the alcian-alizarin stains, the length of the symplectic is difficult to see due to interference at the rostral end by overlap with the palatoquadrate. Consider marking it in some way in the figures.

P9 'We were surprised to observe some mef2ca-associated phenotypes, like shortened symplectic cartilages and fused interhyal joints, were occasionally observed in mef2ca+/+ individuals from the high-penetrance strain (Figure 2B)'

Reader needs to know the frequency with which these phenotypes appeared in wild types.

P9 'we found significant differences between low- and high-penetrance strains for both mef2ca+/+, as well as mef2ca+/- [but homozygous mutants were statistically the same by this measure in the two strains](Figure 2C).'

p10 'the low- and high penetrance strains are both similarly inbred'

They had similar mating schemes, but no data are presented to show that they are similarly inbred. That requires data on heterozygosity. It could be that for one of the strains, for some reason selection was for stronger heterozygosity across the genome than the other strain, that you tended to select for heterozygotes because, for example, inbreeding depresses health subtly in ways that affect the manifestation of the phenotype.

P10 'to generate a 'natural knock down' in mef2ca wild types'

Maybe it doesn't knockdown the mef2ca gene function or expression, maybe it alters the related phenotypes in some other way, in an independent pathway.

P10 'Six *mef2* paralogs in the zebrafish genome share highly conserved amino acid sequences'

Has no one done this type of analysis before that the text could cite? Do the conclusions in this paragraph match the phylogenetic analysis of *Mef2* genes in Chen 2017 *MEF2* signaling and human diseases?

Figure 3. It would be easier for the reader to evaluate the degree of conservation if only conserved amino acids were colored and variants left uncolored.

P11 'To determine the gene expression dynamics of the *mef2* paralogs during craniofacial development'

Consider changing motivation a bit to: 'To determine [whether any *mef2* paralogs have expression domains that overlap mef2ca in the head skeleton], we studied the gene expression dynamics of *mef2* paralogs during craniofacial development'

P11 'and that shared [and divergent] regulatory sequences might control expression'

P11 'expression in isolated wild-type cranial neural crest cells41'

Essential here to tell reader the age of the preparations for scRNA-seq to compare to ages for the other analyses.

scRNA-seq was from isolated neural crest cells, but the work has not established that this is the only cell type that is expressing each of these genes in the head skeleton; if that's true, then it should be mentioned (or maybe I missed it). And the resolution of the scRNA-seq results into anterior arches, frontonasal, posterior arches, and melanocytes seems to come from a method with much less resolution than in situ hybridization to mRNAs would provide.

The selection of crest cells and excluding endodermal and ectodermal cells, which signal to the developing cartilages, removed cells that might very well be contributing to the selected phenotypes because the selection experiments might have altered genes in the signaling pathway that are expressed in these epithelial cell types.

P11 'When we demonstrated that paralog transcriptional adaptation does not account

for the phenotypic differences between the low- and high-penetrance strains34, we did

not examine if selective breeding changed paralog expression between strains in

mef2ca wild types'

Consider change to: 'Data showing that paralog transcriptional adaptation does not account for the phenotypic differences between the low- and high-penetrance strains34 do not rule out the possibility that selective breeding changed paralog expression between strains in mef2ca wild types

P11 'cranial neural crest cells between unselected wild types and high-penetrance wild types'

Were these wild type siblings from an unselected mef2ca mutant line? Or were they from AB or some other wild type line?

Would a better comparison be to compare the two selected lines? Maybe in both the up and the down lines the paralogs are down-regulated with respect to wild types?

P11 'mef2aa, mef2ca, mef2cb and mef2d were all significantly downregulated in the high-penetrance line compared with the unselected strain.'

First, consider change to: 'mef2aa, mef2ca, mef2cb and mef2d were all significantly downregulated in wild-type siblings in the high-penetrance line compared with wild-type siblings from the unselected strain.'

It's unclear why scRNA-seq was used for this analysis instead of a more quantitative approach like the qPCR approach of Figure 4A. It seems like this experiment doesn't utilize the advantages of scRNA-seq and instead uses it for something the method is not well suited for. Visual comparison of the plots requires that the same number of cells are present in each of the two samples. Was that true?

Furthermore, for comparison to Figure 4B, it would be better to show the data in Figure 4C divided into the same four 'cell types'. And even better if shown in the context of all clusters that came out of the experiment.

P11 'all significantly downregulated in high-penetrance heads compared'

Should it read: '…downregulated in the heads of wild-type siblings from high-penetrance lines'? Or were these selected heads that were mutant with high expressivity? I'm just not sure what a 'high penetrance head' is.

P11 'paralogs. We next used RT-qPCR to compare *mef2* paralog expression between highand low-penetrance wild types.'

This experiment would be more interpretable if, like the scRNA-seq experiments, wild types from an unselected line of mef2ca mutants were used. If the unselected line is not in between the two selected lines, then explanations for the results shown would be different than currently in the manuscript.

P12 'generally increased and decreased expression, respectively, of the *mef2* paralogs'

This up and down conclusion can't be drawn if the unselected wild types are not in the middle. But because that control is missing, the conclusion does not seem to be justified, only that the two selected lines differ from each other. While selecting on phenotypes, not expression patterns, both selected lines could be up or both down in paralog expression from the data presented.

Figure S3. Expression of the housekeeping genes was normalized to rps18. Were the *mef2* expression levels normalized to the same gene?

In addition, the conclusion from these data is that there's no difference between low and high lines for these two genes, but no statistical test is given and the standard deviations do not overlap. A statistical test would seem useful, even though the data would seem to show as the authors concluded that the low line is not higher than the high line for these two genes.

P12 'we systematically removed functional copies of mef2ca paralogs from this strain'

This is indeed the best way to address the hypothesis above. The phrase 'from this strain', however, should probably be replaced by 'the low-penetrance strain'.

P12 'is most closely related to'

In what way? In sequence? In expression patterns? In history?

P12 'is the second highest expressed *MEF2* PARALOG in cranial'

P12 'mef2cb homozygous mutants do not have an overt craniofacial Phenotype'

Do they have other phenotypes?

P12 'mef2cb does not function in zebrafish craniofacial development'

Because mef2cb gene, which the sentence means because it uses gene nomenclature, is expressed, then I'd say the mef2cb gene DOES function in craniofacial development. But the Mef2cb protein apparently doesn't provide an essential unique non-redundant function because the mutant has no phenotype.

P12 'when we removed one copy of mef2cb'

Good that the reciprocal experiment was done.

P12 'Removing both copies of mef2cb from mef2ca homozygous mutants IN THE LOW-PENETRANCE STRAIN produces severe'

P12 'there is no difference in symplectic length between wild types and mef2ca heterozygotes (wild type vs. mef2ca+/-;mef2cb+/+)'

Specify genetic background.

P12 'Removing copies of mef2cb from mef2ca wild types does not significantly change'

Again, specify strain/background. Is it the low background in the whole paragraph?

P12 Did the authors also remove mef2cb activity in unselected and high-penetrance lines? If not, then it would be helpful to motivate why choosing one line and not the others.

P12 'We conclude that mef2cb buffers against mef2ca-associated phenotype severity, and

among-individual variation, but not within-individual variation IN THE LOW-PENETRANCE LINE.'

I think that's right, or were all three lines checked? It's hard to tell because the text does not always clearly specify which lines is used.

P13 'mef2d mutant allele in the low-penetrance strain (Figure 6A). Homozygous mef2d mutants did not develop any overt skeletal phenotypes in an otherwise wild-type background'

This is confusing. Is the background the low-penetrance strain, which is not the same as a wild-type background because it has been selected for likely rare alleles in the original background, or is the mef2d mutant in an 'otherwise wild-type background' as the sentence says?

P13 'mef2d, and further increased in double homozygous mutants (Figure 6C)'

This is difficult to follow. This paragraph says that the mef2d mutants were made in the low line, but then Figure 6C indicates it's the unselected strain. Which is correct?

P13 'but mef2ca heterozygotes are more variable when mef2d is disabled (Figure 6E).'

Which phenotype is examined here? The previous sentence makes me think it's only symplectic length variation, but because neither the sentence nor the figure panel nor the figure legend says, it's impossible to know.

P13. 'mef2b is the most divergent mef2ca paralog,'

By what criterion? Sequence? Length? Time at which this variant appeared in phylogeny? Expression pattern?

P14 'Both of these phenotypes are never seen in unselected'

Change to: 'Neither of these phenotypes are ever seen in unselected'

P15 'Our experiments support a model where we selected upon alleles that either amplify or dampen existing vestigial expression'

Are those selected alleles at the mef2ca locus? Or at the *mef2* paralog loci? Or in trans acting genes, presumably transcription factors or their partners, that regulate mef2ca or cb or other genes?

P15 'and is what we selected upon in mef2ca mutants'

This sentence implies that the selection was on variation at the mef2ca mutant locus. Either some mapping data need to be presented (maybe I missed it) that support where the responding loci were in the genome of the selected lines, or some discussion is warranted about where the variation might be in the genome and what types of gene variants might have responded. Maybe this is later in the discussion.

P15 'For example, disabling mef2cb affects most phenotypes'

I think this means 'For example, disabling mef2cb affects most MEF2CA-RELATED CRANIOFACIAL phenotypes.

P15 'primarily affect penetrance of ventral cartilages'

Should be: 'primarily affect penetrance of ventral cartilage PHENOTYPES'.

P15 'Further, expression of all the paralogs was affected by selection, indicating that they all contribute to buffering.'

Not necessarily. It could be that selection acted on a single factor that alters the expression of all of the paralogs in the same way even though the protein of only one or two contributes to the selected phenotype; the others are just along for the ride because of shared upstream regulatory mechanisms, not because of shared downstream functions.

P15 'Closely related paralog expression dynamics are similar'

I guess closely related expression dynamics are of course similar. But I think what the text meant to say was that 'The expression dynamics of paralogs closely related in sequence and phylogeny are similar'

P15 'In this model, few enhancers are inherited that control the expression of many paralogs.'

Again, it would be helpful to reader to know whether the authors think these selected enhancers are in the *mef2* genes themselves or in transcription factors that might regulate coordinately all of the *mef2* paralogs.

P15 'Determining whether the aspect of the mef2ca phenotype buffered by each

paralog is related to their wild type expression pattern'

Consider: 'Determining whether the SPECIFIC PORTION of the mef2ca phenotype THAT EACH PARALOG BUFFERS is related to THAT PARALOG'S wild-type expression pattern'

P16 'phenotyping in this study was limited to the symplectic cartilage.'

Remind reader why this specific element makes the best proxy for the whole system and focusing on it is unlikely to lead to wrong conclusions even if several other phenotypes had been studied additionally.

16 'buffering in our system only regulates among-individual variation, not within-individual variation'

Tell reader why this is an important finding. Why should I care if these two types of variation are regulated by distinct or common mechanisms.

P16 'We propose that cryptic variation in vestigial paralog expression is the noise underlying variable craniofacial development'

Here, come back to the motivation in the beginning about variation in human facial structure. Sceptics might say that, well this doesn't apply to humans because the effects these authors found are mostly related to genes from a fish-specific gene duplication, the a and b copy of mef2c, and because humans just have one MEF2C gene, this is irrelevant. Text could readily disabuse sceptics of this notion and the symmetry of returning to the original motivation would make this more fun.

*Reviewer #2 (Recommendations for the authors):*

1) I found this manuscript difficult to read. In large part, my difficulty was due to finding vague phrases throughout that I would linger over to try to interpret. I will list some of these below, but they were sufficiently numerous as to lessen the impact of this cool work!

Page 1 (the Abstract): "Here we used the zebrafish mef2ca mutant, which produces variable phenotypes, to understand craniofacial variation." This is difficult to understand – is there one mutant or different alleles collectively produce a range of phenotypes.

Page 2: "facial variation persists among genetically homogeneous populations" Does "genetically homogeneous" mean twins or clones? If not, is this useful?

Page 2: "Thus, it is possible that high human facial variability is heritable, and under selection." Do the authors mean that variability is heritable? or that features are heritable and variability of features is under selection?

2) I question whether the study of the penetrance and variability of expression associated with the different mef2ca alleles adds to the study or diverts attention from the take-home messages. In particular, the authors claim that the data presented in Figure 1C shows that mef2caco3008 and mef2cab1086 alleles are associated with different penetrance. This just does not seem to be true from the data – or at least the authors do not indicate what criteria they use to say there is a significant difference.

3) The authors emphasize that a deletion mutation of *mef2* in mammals has a mild phenotype compared with the phenotype of other mutant alleles. Then they sort of suggest that the mef2caco3008 is of particular interest because it is a deletion of the entire coding sequence of mef2ca. I think the strong implication is that the two mutations are equivalent. Nevertheless, the mef2caco3008 zebrafish deletion has a strong phenotype. And then no comment. I think this discussion in the Results section is a distraction. I think too little is known about each mutation to even compare them and suggest they are similar. This is an example of the lack of focus that diminishes the strong points of the manuscript.

4) The authors are not sufficiently careful about their use of and the meanings of the words "up-regulation" and "down-regulation" with respect to paralog expression. One might imagine that selecting for high or low penetrance would select for new mutations that modify expression of the paralogs. Or, more likely in my opinion, that different animals in the starting population have higher or lower expression of the paralogs and the breeding scheme isolates these genomes into lower penetrance or higher penetrance strains. The authors MUST make clear what they mean by these terms. It is not clear in fact whether there ever is a change in gene expression of a paralog. Selection of higher-expressing alleles or lower-expressing alleles from a starting population is not the same as selecting for their up- or down-regulation. The authors really should investigate how much standing variation exists in their populations of fish, especially the unselected starting population.

5) Throughout, the authors make conclusions that are based on assumptions that are not sufficiently supported by data. For example:

On page 6: "transcriptional adaptation is not a major factor underlying phenotypic variation in our system; the PTC (mef2cab1086) allele would be expected to be more mild than the full deletion (mef2caco3008) allele if transcriptional adaptation was a factor." I don't think we fully understand the triggers of transcription adaptation to make this conclusion – especially when it could have been measured directly!

On page 8: "Because the low- and high penetrance strains are both similarly inbred, it is unlikely that there is more genetic variation in the high-penetrance strain background accounting for the increased variation in this strain." This would not be correct if breeding for phenotypic variability turns out to be a selection for genetic variability.

As mentioned above, the authors claim the paralogs have no function in craniofacial development and yet they suggest that diminished expression of the paralogs leads to the expression of mutant phenotypes in the high penetrance strain. They need to clarify and explain.

6) I am concerned that the newly engineered mutations in the paralogs might trigger "transcriptional adaptation" as they each produce premature termination codons – early stops. This would be easy to test.

7) There are lots of places where the authors need to be clearer.

On page 7: the authors discuss the establishment of high penetrance and low penetrance strains – they need to remind us that the mutation is lethal and so they propagate these strains as crosses between heterozygotes (I think?). and they need to remind us or refer to the Methods section to tell us how many generations of selective breeding have taken place.

On page 8: "The zebrafish *mef2* paralogs each encode a MADS box and *MEF2* domain which are remarkably similar (Figure 3B and S1)," – similar to what?

On page 9: "early" and "late" stages of expression should be defined or clarified

8) The entire discussion of "buffering" in the Introduction, etc. is concerning to me. How is the phenotypic buffering that occurs because of vestigial expression of a paralog different than partial "redundancy"? The authors write "To our knowledge, this vestigial buffering hypothesis has not been previously proposed or tested." And yet certainly the work of Yitzhak Pilpel (Nature Genetics 2005; PNAS, 2006) and others very specifically hypothesize that paralogs function as "backup circuits". Further, how does the phenomenon studied here relate to studies of genetic modifiers and what has been learned there. The current work does not operate in isolation, and the Introduction seems to imply that.

*Reviewer #3 (Recommendations for the authors):*

1. A conclusion drawn from the results in Figure 4 is that selecting for high penetrance led to decreased expression of *mef2* paralogs in wild-type embryos, whereas selecting for low penetrance led to increased expression. However, direct comparisons are only made between the unselected vs. high-penetrance strains, and the high-penetrance vs. low-penetrance strains. Without explicitly showing that paralogs are also upregulated in the low-penetrance strain vs. unselected, it can't be fairly concluded that selection for low penetrance worked through this mechanism.

2. The authors note that they removed functional copies of *mef2* paralogs from the low penetrance mef2cab1086 strain, but then used the published fh288 allele for mef2cb. I assume they crossed this line into the low penetrance strain, but for how many generations? The penetrance of the mef2ca phenotypes for this new combination in Figure 5D seem considerably higher than they previously reported (Sucharov et al. 2019), so whether it is still technically "low penetrance" seems questionable. Similarly, the authors initially state that they made the new mef2d and mef2b alleles (co3013, co3012) on the low-penetrance strain (as suggested in the Materials and methods), but then Figure 6C/7C implies that they are assessing phenotypes on an "unselected" strain (the penetrance values listed support that this is not the original low penetrance strain). Please correct this discrepancy. Presumably, the predicted phenotypes and interpretations would be different depending on strain background, though possibly not, given my previous point.

3. Regarding the mild(er) phenotype of the b631 allele: Ensembl and UCSC genome browser list multiple transcripts for mef2ca, some of which have distinct transcriptional and translational start sites. Are any of these alternate forms present in the zebrafish neural crest, and, if so, could they be ameliorating the b631 phenotype? Would all transcripts be impacted by the b1086 mutation?

4. In the discussion on decoupling buffering mechanisms, the authors note that paralogs can share enhancer sequences dating back to before duplication, which is indisputable, but then seem to imply either that each paralog's version of the enhancer has evolved in parallel under selection for high or low penetrance or that a single (or few) enhancer(s) is regulating other paralogs in trans (?). If I am interpreting correctly, more evidence should be provided for each of these claims. This section requires further clarification.

[Editors’ note: further revisions were suggested prior to acceptance, as described below.]

Thank you for resubmitting your work entitled "Variable paralog expression underlies phenotype variation" for further consideration by *eLife*. Your revised article has been evaluated by Christian Landry (Senior Editor) and a Reviewing Editor.

The manuscript is valuable and has been improved but there are some remaining issues that need to be addressed, as outlined below:

The clarity of the text still needs significant further improvement.

– Please make it clear throughout the intro and text that the manuscript consists of two components that are not yet clearly related.

– Discuss that the lack of understanding of how the various alleles affect expression or function of the gene's protein product, and the precise function of the gene in shaping craniofacial structures is not clear. Therefore, it is difficult to derive a mechanistic understanding of the observations and it remains to be investigated if the findings can be extrapolated from this study to other genes.

– Please correct the text according to the issues identified by the two reviewers. The writing is often unclear: terms are used that are poorly defined or incorrectly applied; interpretations are freely mixed with presentations of results; paragraphs often have multiple ideas intermingled leading to a lack of clarity and repetition of ideas in different paragraphs. There are mistakes in the use of some references or the descriptions of some past work. Also descriptions of the precise experimental approach taken should be made clearer in some cases.

*Reviewer #2 (Recommendations for the authors):*

The revised manuscript is still very difficult to read and needs further improvements

This work makes a substantial contribution to our understanding of how genetic background might function to modify the phenotype associated with a particular mutation. Three important take home lessons of this work are:

1) Within populations there is existing (standing) variation in the expression levels of a set of paralogous genes;

2) Control over the expression levels of these paralogues is heritable;

3) Variations in the level of expression of paralogues can modify the expressivity and penetrance associated with a mutant allele of one of the paralogues. The authors use these observations to support a very interesting model they propose in which over evolution as a paralogue loses one of its functions, perhaps by acquiring diminished expression in a particular tissue, it may retain a vestigial capacity to support that function, which may be uncovered by boosting its expression in that tissue. The data presented appear consistent with the hypothesis that the levels of expression of each paralogue may be coordinately regulated, a model that could be readily addressed here but is not directly addressed in this study.

The work is valuable. Its impact is dampened by four factors:

1) The work makes two very different points that are not yet clearly related: the first portion of the manuscript deals with a purported relation between the strength of a mutant allele causing a developmental phenotype and the amount of variability in the structures that are formed abnormally, whereas the second portion of the manuscript deals with the idea that paralogue expression may mitigate or exaggerate the expression of the phenotype associated with a single mutation.

2) The writing is often unclear: terms are used that are poorly defined or incorrectly applied; interpretations are freely mixed with presentations of results; paragraphs often have multiple ideas intermingled leading to a lack of clarity and repetition of ideas in different paragraphs.

3) There are outright mistakes in the use of some references or the descriptions of some past work.

4) Descriptions of the precise experimental approach taken should be made clearer in some cases.

Bailon-Zambrano et al. study factors that influence the expression of phenotype associated with a mutation. In one portion of the work they study a series of alleles of the zebrafish mef2ca gene, which is necessary for normal craniofacial development. They demonstrate that recessive putative loss-of-function mutant alleles of the mef2ca gene of zebrafish are associated with a range of expressivity. By focusing on one aspect of the mutant phenotype, the length of the symplectic cartilages that support the jaw, they find a correlation between the average strength of the phenotype of an allele (measured as reduction in length) and the extent of variability between mutant individuals that carry the allele. This is an interesting idea, but it is not fully explored or proven by their study. Because it is not clear how the various alleles affect expression or function of the gene's protein product, and because the precise function of the gene in shaping craniofacial structures is not clear, it is difficult to derive a mechanistic understanding of their observation and it is not possible to extrapolate from this study to other genes.

The findings that associate individual mef2ca alleles with variability in the severity of the craniofacial phenotype are difficult to interpret. The authors order the "strength" of an allele based on the overall severity of the phenotype, but the molecular reason why the different alleles have different "strengths" is completely unclear. The mildest allele has a mutation affecting the initiation codon, and so presumably it produces a protein utilizing a different initiation point. However it is not clear why the deletion of almost the entire locus leads to an intermediate phenotype, whereas a mutation introducing a premature termination causes the most severe phenotype. Perhaps the severe allele produces a product that interferes with the activity of paralogous products or interacting proteins? Who knows? In addition, since we don't know the precise cellular functions regulated by the gene, it is impossible to generalize from this case and conclude, as the authors do, that there is a general correlation between severity of an allele and the variability in phenotype that results. Finally, this reviewer is concerned about this conclusion and generalizations that may be drawn from focus on a single quantifiable character, the symplectic cartilage. Perhaps there is always a fixed amount of variation in the length of this cartilage. As stronger alleles produce shorter cartilage pieces, similar absolute variations in length may appear to be of greater significance when affecting structures of shorter average length.

In contrast with the first portion of the work, the second portion of the work makes truly valuable and mechanistic contributions to our understanding of how genetic background might modify the phenotypic expression of a particular mutation. The second portion investigates one mechanism that may contribute to the oft-observed phenomenon that a single mutation may be associated with different expressions of a phenotype in a manner that is dependent on the genetic background. The authors focus on loss-of-function of the mef2ca gene of zebrafish, which is needed for the normal development of several craniofacial structures. They find expression levels of *mef2* paralogues may modify the phenotype associated with the mef2ca mutation. Three important take home lessons of this work are:

1) Within populations there is existing (standing) variation in the expression levels of a set of paralogous genes;

2) Control over the expression levels of these paralogues is heritable;

3) Variations in the level of expression of paralogues can modify the expressivity and penetrance associated with a mutant allele of one of the paralogues. The authors use these observations to support a very interesting model they propose in which over evolution as a paralogue loses one of its functions, perhaps by acquiring diminished expression in a particular tissue, it may retain a vestigial capacity to support that function, which may be uncovered by boosting its expression in that tissue.

Major issues:

1) I think the work is important, I would like to see it published, but I found the manuscript very difficult to read.

2) Perhaps one reason it is difficult to follow is that the work described here is composed of two very different studies, one describing the phenotypic effects of three different alleles and the other describing the effects of genetic background on the expression of one allele. This is reflected in the presence of two independent sections in the Introduction. Because we don't understand how the three different mutations affect generation of a protein product, or how they might affect the expression of paralogues (which are shown to be very important in the second half of this work), we cannot understand what a "strong" or "intermediate" allele is. IF the deletion completely removes gene function, then from a molecular point of view, it is a complete loss of function, and thus the premature termination mutation is a "special" type of allele that is more than simply completely loss of function. In any case, without insight into how these alleles work, I don't think it is fair to make generalizations about strengths of alleles and variability in the expression of a phenotype.

On the other hand, there are clear statements in the Introduction that could really be used as the crux of an entire paper: "whether paralog expression variation underlies phenotypic variation has not been directly tested. Finally, whether variation paralogous buffering is subject to selection is unknown." Personally, I suggest focusing on the effects of paralogues to buffer phenotype – it makes for a crisp introduction and focused story.

3) There are assertions that I believe are incorrect.

For example, lines 110-113: "Because mutations in none of the other four zebrafish paralogous *mef2* genes have been reported, their function is unknown. These paralogs have human orthologs, MEF2A, MEF2B and MEF2D, that do not appear to be critical for craniofacial development46-48." Reference 46 and perhaps the others does NOT address the question of whether these genes are required for craniofacial development – ref 46 says mef2d fusion proteins are associated with cancer. These are somatic gain-of-function mutations and irrelevant to the matter at hand, which is complete whole animal loss of function.

Another example, lines 307-311: "Data showing that paralog transcriptional adaptation does not account for the phenotypic differences between the low- and high-penetrance strains34 do not rule out the possibility that selective breeding changed paralog expression between strains in mef2ca wild types." Reference 34 is the classic Force et al. paper and is not the correct reference – I think you meant to use reference 52.

Another example, lines 364-367: "We detected mef2d transcripts in the anterior arch population of 24 hpf wild-type cranial neural crest cells, and mef2d expression is significantly lower in the high-penetrance strain compared with the high-penetrance strain" I think you meant to compare high and low penetrance strains.

There are other examples of assertions that are a bit glib and that distract from the strength of the work.

4) Terminology is used loosely or incorrectly.

I find in the literature the use of between-individual as well as among-individual, and that between-individual is used more often. Personally I find between-individual easier to keep in my mind what the comparison being described is, but both are legit I guess.

In many cases I believe it would be clearer to say that there are differences in expression levels among members of a population and that these levels are heritable – rather than saying "variability is heritable" (line 42 and elsewhere), which is a difficult concept to grasp precisely.

Lines 17-18: "Craniofacial disorders further increase this variability." "this variability" is ill-defined – perhaps it is more useful to speak of "range of variability"

Line 44: "Finish people" should be "Finnish people".

Lines 63-64: "Human genetic craniofacial disease phenotypes also appear more variable than normal human facial phenotypes." What is the actual comparison that is being made? I can't reconstruct this experiment.

Lines 70-71: "Patients heterozygous for mutations affecting the transcription factor encoding gene MEF2C show variable facial dysmorphologies19-21." This is relevant to the present study if you are describing variability associated with a single mutation – if that is true, make it explicit. If you mean that there are different alleles and these different alleles generate a range of phenotypes then it is not relevant to the discussion here.

Line 187 – not a "range of alleles", but a series of alleles.

Line 225-227 and elsewhere – there are lots of places where alleles are described as variable – where the mutation or gene is conflated with the phenotype. "and the most severe mutant (mef2cab1086) allele was significantly more variable than the mildest mutant (mef2cab631) allele".

Line 266: "likely to be robust against null mutations" this is not clear.

The entire section beginning with line 284: "Closely related *mef2* paralogs share similar expression dynamics" It is not clear what conclusions can be made or what conclusions you want to make about similar "dynamics". What one would like to measure is whether the paralogues are expressed in the same cells as the gene of interest and at a time when they are performing a function of interest. Since you have not explained when and where mef2ca functions, it is impossible to interpret the "dynamics" data in terms of relevant function. Please be explicit about what these experiments let you conclude and why.

Line 296-297: this paragraph is ended with a summary sentence: "Of note, previous work indicates that partially shared regulatory motifs are predictive of robustness66." Neither the meaning of this nor its relevance is clear to me. whose robustness? And in what context?

5) Some things can just be made simpler:

Lines 32-34: "We present a novel, mechanistic model for phenotypic variation where cryptically variable, vestigial paralog expression buffers development." I don't think the Reader will understand "cryptically variable" at this point of the paper (the Abstract).

Line 61 and elsewhere: "genetic mutations" could be simply "mutations".

6) Experiments and Models that need to made clearer:

A critical series of experiments involves study of mutant alleles of the paralogues that are thought to be insufficient to have a dramatic effect on the craniofacial phenotype, but are thought to modify this phenotype in the context of mef2ca or other paralogue mutations. The authors at one point talk about this experiment as simply "removing" the paralogue (line 343). At that point, I had assumed the new mutations were being generated within the strain that was being studied – that is the easiest way to interpret "remove". Instead, paralogue mutations are induced elsewhere and bred into strains for study. The rationale for the experimental design is complicated and should be explicitly set forward. I would ask the authors to make clearer within the Results section how they generated the new mutations in the paralogues and how these were introduced into various strains and what precisely was examined. The entire exposition is a bit buried in Lines 336 – 338 paralogs. "For all paralog functional experiments, we introduced the relevant paralog mutation into the low-penetrance strain, then maintained by outcrossing to unselected AB." This is not a complete sentence. The details of this experiment are really important.

The other point that needs to be better explained and clarified is their model in Figure 9. The mechanistic implications of the illustration are vague, but to me there appears to be focus on the roles of enhancers – the figure seems to imply (i) that enhancers acquire mutations that diminish expression of the paralogue and (ii) that under selection the enhancer acquires mutations that result in increased expression of the gene. I would think rather that standing variation arises prior to the selection – and this might be illustrated in the figure. Nevertheless, it seems from the data that the expression levels of all the paralogues may be coordinately regulated – and this idea seems to supported in the Discussion. If all the paralogues are expressed at low levels in the high penetrance strain and at higher levels in the low penetrance strain, then it is more likely that variation is not being driven at the level of each paralogue's enhancer but rather by some factor working in trans. That idea is really missing from Figure 9 – and since it is a summary illustration of what the authors think, getting this Figure right is important. In truth, I would have liked to see a few crosses between strains to test the idea that a single modifier working in trans is responsible for all the effects – THAT would have been both relatively easy to test and killer interesting as a result. But without such an experiment, the authors need to be very clear about alternate possible interpretations.

*Reviewer #3 (Recommendations for the authors):*

The authors have extensively revised and added new data to their manuscript in response to the initial round of reviews. The language is generally more precise and concise, and sections lacking clarity are improved or have been removed. I especially appreciate that they now explain the breeding scheme by which the paralog mutant alleles were established and maintained, as the phenotype prevalences are now better in line with what the reader is led to predict.

---

## [Author Response]

Essential revisions:The three reviewers agree that the question addressed is of great interest to evolutionary and developmental biologists in general and to those studying the evolution of developmental mechanisms in particular. The manuscript reports substantial and interesting advances, but in its current form it is difficult to read, which led to significant frustration among the reviewers. The manuscript needs to be substantially rewritten and shortened. The writing and interpretation of the results needs to be clarified as many statements are perceived as lacking precision. One of many examples is the statement that each selected strain has similar inbredness. However, there is no evidence that this was actually tested. The authors should also ensure that their conclusions match the results and are not over interpreted, pointing out explicitly caveats where necessary. Finally, the results need to be discussed in relation to what has been published, for example how does the phenomenon studied here relate to studies of genetic modifiers and what has been learned there.

We are thankful that the reviewers and editors appreciated the “substantial and interesting advances” reported in this work. Our revised manuscript contains important new data further supporting our conclusions, and advancing our model. Furthermore, the text has been extensively rewritten for clarity and precision. Thus, we have fully addressed all the reviewer critiques. See below for a point-by-point response.

Reviewer #1 (Recommendations for the authors):P3 abstract: 'In support, mutagenizing all mef2ca paralogs in the low penetrance strain'But one wasn't mutated. It would be helpful to say which paralogs were mutated in the abstract just so that literature searches by people studying the paralogs would be more likely to find this paper.P3 'Human craniofacial variation allows us – and even computers -- to identify each other.

Done

Re: 'facial variation persists among genetically homogeneous populations1.Text should specifically talk about identical twins here.

Done

P4 paragraph should start with a topic sentence: 'Variation can be talked about in two different ways: penetrance is….'

Done

P5 ‘large deletion encompassing the MEF2C locus’Is that in homozygous or heterozygous condition?

Done

P5 text says that mef2cb mutants are viable without craniofacial phenotypes, but do they have other phenotypes?

Done

P6 'Because no other zebrafish mef2 gene mutants'. Reader needs to know how many mef2 genes zebrafish has and their relationship to human MEF2 genes.

Done

P6 'nearly all mef2ca associated phenotypes changed penetrance as well34'Did they change in the same direction? All more severe or all less severe?

Done

P6 'the spread in symplectic length'Define what this 'spread' means in phenotypic terms.

Done

P6 'linear measurements of the symplectic cartilage, allowing us to measure severity,among-individual variation, and within-individual variation in the zebrafish craniofacial skeleton'Authors need to demonstrate that measuring the length of this single skeletal element is sufficient to capture the variation they intend to study.

This is an excellent suggestion, and we now include data to indicate that shorter symplectic cartilage length is correlated with expanded opercle bone suggesting that the symplectic is a representative phenotype for the other craniofacial phenotypes affected in *mef2ca* mutants (Figure 1—figure supplement 2).

P7 'upregulated in the low penetrance strain compared with the high-penetrance strain'With respect to what internal control?

Done

P7 'partially due to individuals inheriting different mutant alleles retaining different levelsof functional activity'Right, for protein function, but also for different transcriptional expression domains or levels.

Done

P7 'The C-terminal domain is more divergent among different mef2 genes38.'This paragraph is talking about alleles of one gene but this sentence is talking about different genes, so it's a bit confusing and seems out of place.

We deleted this statement.

P7 'destroys the initiating methionine35.'Is any protein made from an internal methionine that might provide an alternative initiation site?

Thank you for this question, to address this we performed a new experiment. We now include antibody stains for both the *mef2ca* methionine allele (b631) and the deletion allele (co3008) these results further our understanding of the different alleles. (Figure 1—figure supplement 1).

P8 'deletion allele found in a [mildly affected] patient23'

Done

P8 'among-individual variation associated with the opercle bone phenotype; one individual has phenotypically wild-type opercles while the other individual has bilateral mutant phenotypes'Give each of the individuals in Figure 1B a different panel number, like B1, B2 for the two wild types. Then in the sentence above call them out by name so the reader doesn't have to figure out which is which.

We thank the reviewer for this helpful comment, we made the suggested changes that improved clarity.

P8 'within-individual (left-right) opercle phenotype variation is present in one of the mef2cab631 animals (lower)Be a bit more specific, saying that in the bottom individual (which could be called B4), the left opercle is relatively normal but the right opercle is partially duplicated (or something like that).Figure 1D. I think it would be helpful for the reader to label all of the skeletal elements in Figure 1D and highlight the symplectic by color.

This helps clarity, thank you for the suggestion.

P8 'Ordering our allelic series by expressivity [of this one element] shows'What about the other phenotypes? Do they follow the same allelic series? It's possible that different alleles could affect differentially different phenotypes.

Done

P8 'expected to be [milder] than the full deletion'P8 'the PTC (mef2cab1086) allele would be expected to be more mild than the full deletion(mef2caco3008) allele if transcriptional adaptation was a factor.'Yes, agreed.P9 'the most severe allele is also the most variable (Figure 1F).'Consider plotting a measure of severity on the vertical axis at the right of the figure on the same horizontal axis to make the point about the correlation.

This is a good idea, but because we have already ordered by severity in this graph, the suggested change is not likely to affect the interpretation. Furthermore, we clarified in the text and figure legend that the x-axis is ordered by our previously determined severity by both penetrance and expressivity.

P9 'there is no significant difference [among] mutant alleles for this type of variation '

Done

P9 'Severity is positively correlated with variation when comparing'Would it make more sense to say 'Variation is positively correlated with severity when comparing'? Both of course are correct but phrase below says it in this order and biologically it seems to make more sense to me that the severity is a property of the allele and that the variation is a property of the biology arising from the altered protein function.

We fully agree and changed as suggested.

P9 'we hypothesized that [a single] mutant allele, mef2cab1086,'

Done

P9 'shortened symplectic cartilages and fused interhyal joints, were occasionally observed'In nearly all of the alcian-alizarin stains, the length of the symplectic is difficult to see due to interference at the rostral end by overlap with the palatoquadrate. Consider marking it in some way in the figures.

This is a good suggestion, we now outlined the symplectic cartilage in all figures containing Alcian-Alizarin stains.

P9 'We were surprised to observe some mef2ca-associated phenotypes, like shortened symplectic cartilages and fused interhyal joints, were occasionally observed in mef2ca+/+ individuals from the high-penetrance strain (Figure 2B)'

Done

Reader needs to know the frequency with which these phenotypes appeared in wild types.P9 'we found significant differences between low- and high-penetrance strains for both mef2ca+/+, as well as mef2ca+/- [but homozygous mutants were statistically the same by this measure in the two strains](Figure 2C).'

Done

p10 'the low- and high penetrance strains are both similarly inbred'They had similar mating schemes, but no data are presented to show that they are similarly inbred. That requires data on heterozygosity. It could be that for one of the strains, for some reason selection was for stronger heterozygosity across the genome than the other strain, that you tended to select for heterozygotes because, for example, inbreeding depresses health subtly in ways that affect the manifestation of the phenotype.

Agree, we removed all statements asserting that the strains are similarly inbred.

P10 'to generate a 'natural knock down' in mef2ca wild types'Maybe it doesn't knockdown the mef2ca gene function or expression, maybe it alters the related phenotypes in some other way, in an independent pathway.

We removed this statement.

P10 'Six mef2 paralogs in the zebrafish genome share highly conserved amino acid sequences'Has no one done this type of analysis before that the text could cite? Do the conclusions in this paragraph match the phylogenetic analysis of Mef2 genes in Chen 2017 MEF2 signaling and human diseases?

Thank you for this suggestion, we now reference several studies that performed phylogenetic analyses on *mef2* paralogs and orthologs. We also point out that we arrived at similar conclusions as these other studies.

Figure 3. It would be easier for the reader to evaluate the degree of conservation if only conserved amino acids were colored and variants left uncolored.

Done

P11 'To determine the gene expression dynamics of the mef2 paralogs during craniofacial development'Consider changing motivation a bit to: 'To determine [whether any mef2 paralogs have expression domains that overlap mef2ca in the head skeleton], we studied the gene expression dynamics of mef2 paralogs during craniofacial development'

Done

P11 'and that shared [and divergent] regulatory sequences might control expression'

Done

P11 'expression in isolated wild-type cranial neural crest cells41'Essential here to tell reader the age of the preparations for scRNA-seq to compare to ages for the other analyses.

Done

scRNA-seq was from isolated neural crest cells, but the work has not established that this is the only cell type that is expressing each of these genes in the head skeleton; if that's true, then it should be mentioned (or maybe I missed it). And the resolution of the scRNA-seq results into anterior arches, frontonasal, posterior arches, and melanocytes seems to come from a method with much less resolution than in situ hybridization to mRNAs would provide.

We agree with this comment. In response, we clarify in the text that sorting for neural crest cells does not capture all the cells involved in craniofacial development. But does allow us to focus on the anterior arches. Furthermore, in response to this comment we removed the sorted single cell RNA sequencing experiment comparing paralog expression between the unselected and high-penetrance strains. We now only include the qPCR comparison between high and low penetrance strains. We still include single-cell RNAseq from isolated NCC data from the unselected strain to demonstrate the relative expression of the *mef2* paralogs in these bone and cartilage progenitor cells.

The selection of crest cells and excluding endodermal and ectodermal cells, which signal to the developing cartilages, removed cells that might very well be contributing to the selected phenotypes because the selection experiments might have altered genes in the signaling pathway that are expressed in these epithelial cell types.P11 'When we demonstrated that paralog transcriptional adaptation does not accountfor the phenotypic differences between the low- and high-penetrance strains34, we didnot examine if selective breeding changed paralog expression between strains inmef2ca wild types'Consider change to: 'Data showing that paralog transcriptional adaptation does not account for the phenotypic differences between the low- and high-penetrance strains34 do not rule out the possibility that selective breeding changed paralog expression between strains in mef2ca wild types

Done

P11 'cranial neural crest cells between unselected wild types and high-penetrance wild types'Were these wild type siblings from an unselected mef2ca mutant line? Or were they from AB or some other wild type line?Would a better comparison be to compare the two selected lines? Maybe in both the up and the down lines the paralogs are down-regulated with respect to wild types?

We removed these data comparing unselected to high-penetrance with single cell RNA sequencing.

P11 'mef2aa, mef2ca, mef2cb and mef2d were all significantly downregulated in the high-penetrance line compared with the unselected strain.'First, consider change to: 'mef2aa, mef2ca, mef2cb and mef2d were all significantly downregulated in wild-type siblings in the high-penetrance line compared with wild-type siblings from the unselected strain.'

Done

It's unclear why scRNA-seq was used for this analysis instead of a more quantitative approach like the qPCR approach of Figure 4A. It seems like this experiment doesn't utilize the advantages of scRNA-seq and instead uses it for something the method is not well suited for. Visual comparison of the plots requires that the same number of cells are present in each of the two samples. Was that true?

In response to this comment, we removed these analyses from the manuscript.

Furthermore, for comparison to Figure 4B, it would be better to show the data in Figure 4C divided into the same four 'cell types'. And even better if shown in the context of all clusters that came out of the experiment.

See above, these analyses were removed.

P11 'all significantly downregulated in high-penetrance heads compared'Should it read: '…downregulated in the heads of wild-type siblings from high-penetrance lines'? Or were these selected heads that were mutant with high expressivity? I'm just not sure what a 'high penetrance head' is.

We edited this sentence for clarity.

P11 'paralogs. We next used RT-qPCR to compare mef2 paralog expression between highand low-penetrance wild types.'This experiment would be more interpretable if, like the scRNA-seq experiments, wild types from an unselected line of mef2ca mutants were used. If the unselected line is not in between the two selected lines, then explanations for the results shown would be different than currently in the manuscript.

This is an excellent suggestion. In response, we performed this key experiment which considerably improved our model. We discovered significant differences between unselected AB strain families (Figure 4D). These important new data motivate the following interpretations: unselected AB strains exhibit standing variation in paralog expression, and this variation is likely what we selected upon. We are extremely thankful for this suggestion which considerably improved our understanding of the system.

P12 'generally increased and decreased expression, respectively, of the mef2 paralogs'This up and down conclusion can't be drawn if the unselected wild types are not in the middle. But because that control is missing, the conclusion does not seem to be justified, only that the two selected lines differ from each other. While selecting on phenotypes, not expression patterns, both selected lines could be up or both down in paralog expression from the data presented.

We agree and changed the text to reflect our new unselected qPCR expression data (see above).

Figure S3. Expression of the housekeeping genes was normalized to rps18. Were the mef2 expression levels normalized to the same gene?

Yes, we used the same normalization control (*rps18*). And your attention to detail motivated us to run a t-test finding that expression is significantly higher in the high penetrance strain. We present both these facts in the figure legend. These results do not alter our conclusions.

In addition, the conclusion from these data is that there's no difference between low and high lines for these two genes, but no statistical test is given and the standard deviations do not overlap. A statistical test would seem useful, even though the data would seem to show as the authors concluded that the low line is not higher than the high line for these two genes.

Done, see above

P12 'we systematically removed functional copies of mef2ca paralogs from this strain'This is indeed the best way to address the hypothesis above. The phrase 'from this strain', however, should probably be replaced by 'the low-penetrance strain'.

See methods, we clarify that the new CRISPR alleles were originally generated in the low-penetrance strain but then were outcrossed to the unselected AB background for several generations. We now consider these to be unselected and changed the text throughout accordingly.

P12 'is most closely related to'In what way? In sequence? In expression patterns? In history?

Sequence similarity, we now clarify this in the text.

P12 'is the second highest expressed MEF2 PARALOG in cranial'

Done

P12 'mef2cb homozygous mutants do not have an overt craniofacial Phenotype'Do they have other phenotypes?

We now address this in the introduction

P12 'mef2cb does not function in zebrafish craniofacial development'Because mef2cb gene, which the sentence means because it uses gene nomenclature, is expressed, then I'd say the mef2cb gene DOES function in craniofacial development. But the Mef2cb protein apparently doesn't provide an essential unique non-redundant function because the mutant has no phenotype.

We agree and changed the wording to be more precise and specific.

P12 'when we removed one copy of mef2cb'Good that the reciprocal experiment was done.

Thanks

P12 'Removing both copies of mef2cb from mef2ca homozygous mutants IN THE LOW-PENETRANCE STRAIN produces severe'

We changed our description of the maintenance of these lines to clarify that this is no longer true breeding low penetrance strain.

P12 'there is no difference in symplectic length between wild types and mef2ca heterozygotes (wild type vs. mef2ca+/-;mef2cb+/+)'Specify genetic background.

We clarify these are unselected, see above.

P12 'Removing copies of mef2cb from mef2ca wild types does not significantly change'Again, specify strain/background. Is it the low background in the whole paragraph?P12 Did the authors also remove mef2cb activity in unselected and high-penetrance lines? If not, then it would be helpful to motivate why choosing one line and not the others.

We clarify these are unselected, see above.

P12 'We conclude that mef2cb buffers against mef2ca-associated phenotype severity, andamong-individual variation, but not within-individual variation IN THE LOW-PENETRANCE LINE.'I think that's right, or were all three lines checked? It's hard to tell because the text does not always clearly specify which lines is used.

We clarify these are unselected, see above.

P13 'mef2d mutant allele in the low-penetrance strain (Figure 6A). Homozygous mef2d mutants did not develop any overt skeletal phenotypes in an otherwise wild-type background'This is confusing. Is the background the low-penetrance strain, which is not the same as a wild-type background because it has been selected for likely rare alleles in the original background, or is the mef2d mutant in an 'otherwise wild-type background' as the sentence says?

See above, we clarify in the text that we consider these unselected.

P13 'mef2d, and further increased in double homozygous mutants (Figure 6C)'This is difficult to follow. This paragraph says that the mef2d mutants were made in the low line, but then Figure 6C indicates it's the unselected strain. Which is correct?

See above, we clarify in the text that we consider these unselected.

P13 'but mef2ca heterozygotes are more variable when mef2d is disabled (Figure 6E).'Which phenotype is examined here? The previous sentence makes me think it's only symplectic length variation, but because neither the sentence nor the figure panel nor the figure legend says, it's impossible to know.

We clarified this in the text.

P13. 'mef2b is the most divergent mef2ca paralog,'By what criterion? Sequence? Length? Time at which this variant appeared in phylogeny? Expression pattern?

We clarify that this is by sequence analyses. This finding is consistent with several other previous studies, which we now cite.

P14 'Both of these phenotypes are never seen in unselected'Change to: 'Neither of these phenotypes are ever seen in unselected'

Done

P15 'Our experiments support a model where we selected upon alleles that either amplify or dampen existing vestigial expression'Are those selected alleles at the mef2ca locus? Or at the mef2 paralog loci? Or in trans acting genes, presumably transcription factors or their partners, that regulate mef2ca or cb or other genes?

Done

P15 'and is what we selected upon in mef2ca mutants'This sentence implies that the selection was on variation at the mef2ca mutant locus. Either some mapping data need to be presented (maybe I missed it) that support where the responding loci were in the genome of the selected lines, or some discussion is warranted about where the variation might be in the genome and what types of gene variants might have responded. Maybe this is later in the discussion.

We now include a considerable discussion about where the variation might be in the genome.

P15 'For example, disabling mef2cb affects most phenotypes'I think this means 'For example, disabling mef2cb affects most MEF2CA-RELATED CRANIOFACIAL phenotypes.

Done

P15 'primarily affect penetrance of ventral cartilages'Should be: 'primarily affect penetrance of ventral cartilage PHENOTYPES'.

Done

P15 'Further, expression of all the paralogs was affected by selection, indicating that they all contribute to buffering.'Not necessarily. It could be that selection acted on a single factor that alters the expression of all of the paralogs in the same way even though the protein of only one or two contributes to the selected phenotype; the others are just along for the ride because of shared upstream regulatory mechanisms, not because of shared downstream functions.

We agree and extensively edited this section.

P15 'Closely related paralog expression dynamics are similar'I guess closely related expression dynamics are of course similar. But I think what the text meant to say was that 'The expression dynamics of paralogs closely related in sequence and phylogeny are similar'

We agree and extensively edited this section.

P15 'In this model, few enhancers are inherited that control the expression of many paralogs.'Again, it would be helpful to reader to know whether the authors think these selected enhancers are in the mef2 genes themselves or in transcription factors that might regulate coordinately all of the mef2 paralogs.

We address this in the extensively rewritten discussion.

P15 'Determining whether the aspect of the mef2ca phenotype buffered by eachparalog is related to their wild type expression pattern'Consider: 'Determining whether the SPECIFIC PORTION of the mef2ca phenotype THAT EACH PARALOG BUFFERS is related to THAT PARALOG'S wild-type expression pattern'

Done

P16 'phenotyping in this study was limited to the symplectic cartilage.'Remind reader why this specific element makes the best proxy for the whole system and focusing on it is unlikely to lead to wrong conclusions even if several other phenotypes had been studied additionally.

Done

16 'buffering in our system only regulates among-individual variation, not within-individual variation'Tell reader why this is an important finding. Why should I care if these two types of variation are regulated by distinct or common mechanisms.

Done

P16 'We propose that cryptic variation in vestigial paralog expression is the noise underlying variable craniofacial development'Here, come back to the motivation in the beginning about variation in human facial structure. Sceptics might say that, well this doesn't apply to humans because the effects these authors found are mostly related to genes from a fish-specific gene duplication, the a and b copy of mef2c, and because humans just have one MEF2C gene, this is irrelevant. Text could readily disabuse sceptics of this notion and the symmetry of returning to the original motivation would make this more fun.

Done, excellent suggestion thank you.

Reviewer #2 (Recommendations for the authors):Major issues:1) I found this manuscript difficult to read. In large part, my difficulty was due to finding vague phrases throughout that I would linger over to try to interpret. I will list some of these below, but they were sufficiently numerous as to lessen the impact of this cool work!

Thank you for appreciating our work. Your comments and those of the other reviewers led us to considerably revise the manuscript striving for clarity.

Page 1 (the Abstract): "Here we used the zebrafish mef2ca mutant, which produces variable phenotypes, to understand craniofacial variation." This is difficult to understand – is there one mutant or different alleles collectively produce a range of phenotypes.

We clarified this section.

Page 2: "facial variation persists among genetically homogeneous populations" Does "genetically homogeneous" mean twins or clones? If not, is this useful?

We clarified this.

Page 2: "Thus, it is possible that high human facial variability is heritable, and under selection." Do the authors mean that variability is heritable? or that features are heritable and variability of features is under selection?

We clarified this language.

2) I question whether the study of the penetrance and variability of expression associated with the different mef2ca alleles adds to the study or diverts attention from the take-home messages. In particular, the authors claim that the data presented in Figure 1C shows that mef2caco3008 and mef2cab1086 alleles are associated with different penetrance. This just does not seem to be true from the data – or at least the authors do not indicate what criteria they use to say there is a significant difference.

By our statistical analyses, the b1086 allele has a higher penetrance of ectopic op and ih joint fusion (Figure 1C) compared with b631. In contrast penetrance of these phenotypes in the co3008 allele is not significantly different from the mildest b631 allele. Furthermore, the total sy length (Figure 1E) is significantly shorter in b1086 compared with co3008. These lines of evidence indicate that b1086 is more severe than co3008. This is clarified in the text.

3) The authors emphasize that a deletion mutation of mef2 in mammals has a mild phenotype compared with the phenotype of other mutant alleles. Then they sort of suggest that the mef2caco3008 is of particular interest because it is a deletion of the entire coding sequence of mef2ca. I think the strong implication is that the two mutations are equivalent. Nevertheless, the mef2caco3008 zebrafish deletion has a strong phenotype. And then no comment. I think this discussion in the Results section is a distraction. I think too little is known about each mutation to even compare them and suggest they are similar. This is an example of the lack of focus that diminishes the strong points of the manuscript.

We agree and changed the language describing this allele indicating that it approximates the human deletion allele. Furthermore, we include new data addressing the protein produced from these alleles which increase our understanding of each mutation.

4) The authors are not sufficiently careful about their use of and the meanings of the words "up-regulation" and "down-regulation" with respect to paralog expression. One might imagine that selecting for high or low penetrance would select for new mutations that modify expression of the paralogs. Or, more likely in my opinion, that different animals in the starting population have higher or lower expression of the paralogs and the breeding scheme isolates these genomes into lower penetrance or higher penetrance strains. The authors MUST make clear what they mean by these terms. It is not clear in fact whether there ever is a change in gene expression of a paralog. Selection of higher-expressing alleles or lower-expressing alleles from a starting population is not the same as selecting for their up- or down-regulation. The authors really should investigate how much standing variation exists in their populations of fish, especially the unselected starting population.

This reviewer proposes the excellent hypothesis that “different animals in the starting population have higher or lower expression of the paralogs and the breeding scheme isolates these genomes into lower penetrance or higher penetrance strains.” In response we directly tested this model and found that there is significant standing variation in paralog expression in the unselected strains (Figure 4D). We would like to thank this reviewer for motivating this very important experiment that dramatically improved the manuscript and clarified our model. This amended model is now clearly conveyed in the text.

5) Throughout, the authors make conclusions that are based on assumptions that are not sufficiently supported by data. For example:On page 6: "transcriptional adaptation is not a major factor underlying phenotypic variation in our system; the PTC (mef2cab1086) allele would be expected to be more mild than the full deletion (mef2caco3008) allele if transcriptional adaptation was a factor." I don't think we fully understand the triggers of transcription adaptation to make this conclusion – especially when it could have been measured directly!

We disagree with this reviewer comment. Our previous work (Sucharov 2019) extensively tested the hypothesis that differences in transcriptional adaptation account for differences in our selectively bred strains finding that it is not a major factor in our system. Nevertheless, we qualify this statement in the text by stating that we did not directly test if the deletion allele induces transcriptional adaptation.

On page 8: "Because the low- and high penetrance strains are both similarly inbred, it is unlikely that there is more genetic variation in the high-penetrance strain background accounting for the increased variation in this strain." This would not be correct if breeding for phenotypic variability turns out to be a selection for genetic variability.

We agree with this comment and edited the text accordingly.

As mentioned above, the authors claim the paralogs have no function in craniofacial development and yet they suggest that diminished expression of the paralogs leads to the expression of mutant phenotypes in the high penetrance strain. They need to clarify and explain.

We agree with this comment and edited the text accordingly.

6) I am concerned that the newly engineered mutations in the paralogs might trigger "transcriptional adaptation" as they each produce premature termination codons – early stops. This would be easy to test.

This reviewer is correct, the new paralog mutations might trigger transcriptional adaptation. However, determining if they do is beyond the scope of this work, and not critical to the conclusions we draw. Future studies will test this hypothesis.

7) There are lots of places where the authors need to be clearer.On page 7: the authors discuss the establishment of high penetrance and low penetrance strains – they need to remind us that the mutation is lethal and so they propagate these strains as crosses between heterozygotes (I think?). and they need to remind us or refer to the Methods section to tell us how many generations of selective breeding have taken place.

We have gone to great lengths to make this manuscript clearer, including the breeding methods description. Specifically, we added a brief description of it to the methods and figure legend of this manuscript including how many generations of selective breeding have taken place. Furthermore, this method is clearly described in three previous manuscripts (Nichols 2016, Brooks 2017, Sucharov 2019) which we reference.

On page 8: "The zebrafish mef2 paralogs each encode a MADS box and MEF2 domain which are remarkably similar (Figure 3B and S1)," – similar to what?

Done

On page 9: "early" and "late" stages of expression should be defined or clarified

Done

8) The entire discussion of "buffering" in the Introduction, etc. is concerning to me. How is the phenotypic buffering that occurs because of vestigial expression of a paralog different than partial "redundancy"? The authors write "To our knowledge, this vestigial buffering hypothesis has not been previously proposed or tested." And yet certainly the work of Yitzhak Pilpel (Nature Genetics 2005; PNAS, 2006) and others very specifically hypothesize that paralogs function as "backup circuits". Further, how does the phenomenon studied here relate to studies of genetic modifiers and what has been learned there. The current work does not operate in isolation, and the Introduction seems to imply that.

Thank you for this comment. In response, we clarified our language throughout and included numerous new citations referencing many of the important manuscripts that set the stage for our work. Our study is placed in much better context following this suggestion.

Reviewer #3 (Recommendations for the authors):1. A conclusion drawn from the results in Figure 4 is that selecting for high penetrance led to decreased expression of mef2 paralogs in wild-type embryos, whereas selecting for low penetrance led to increased expression. However, direct comparisons are only made between the unselected vs. high-penetrance strains, and the high-penetrance vs. low-penetrance strains. Without explicitly showing that paralogs are also upregulated in the low-penetrance strain vs. unselected, it can't be fairly concluded that selection for low penetrance worked through this mechanism.

We agree, our new unselected AB paralog expression study (Figure 4D) strengthens our conclusion (see above).

2. The authors note that they removed functional copies of mef2 paralogs from the low penetrance mef2cab1086 strain, but then used the published fh288 allele for mef2cb. I assume they crossed this line into the low penetrance strain, but for how many generations? The penetrance of the mef2ca phenotypes for this new combination in Figure 5D seem considerably higher than they previously reported (Sucharov et al. 2019), so whether it is still technically "low penetrance" seems questionable. Similarly, the authors initially state that they made the new mef2d and mef2b alleles (co3013, co3012) on the low-penetrance strain (as suggested in the Materials and methods), but then Figure 6C/7C implies that they are assessing phenotypes on an "unselected" strain (the penetrance values listed support that this is not the original low penetrance strain). Please correct this discrepancy. Presumably, the predicted phenotypes and interpretations would be different depending on strain background, though possibly not, given my previous point.

We agree with these points. We now clarify that the fh288 allele was crossed to low penetrance which is, as you state, no longer low penetrance. We then maintained by outcrossing to AB. Thus, we consider it to be unselected. Similarly, the new CRISPR mutants were originally created on the low-penetrance strain, but maintained by outcrossing these alleles to unselected AB. We also consider them to be unselected. We now clearly describe our allele generation and maintenance strategy in the methods and text.

3. Regarding the mild(er) phenotype of the b631 allele: Ensembl and UCSC genome browser list multiple transcripts for mef2ca, some of which have distinct transcriptional and translational start sites. Are any of these alternate forms present in the zebrafish neural crest, and, if so, could they be ameliorating the b631 phenotype? Would all transcripts be impacted by the b1086 mutation?

Thank you for this point. We include new Mef2c protein (immunostaining) data indicating that protein is produced from the b631 allele, as this reviewer hypothesizes (Figure 1—figure supplement 1). The protein we detect is likely from the downstream in frame ATGs that the reviewer mentions. Future protein study (Western blot) will be informative. We also include new protein data with the deletion allele. Together with our previous report (Sucharov 2019), these data indicate that the two more severe alleles (b1086 and co3008) do not produce meaningful amount of Mef2ca protein but b631 does. We incorporated these new data and interpretations into the text.

4. In the discussion on decoupling buffering mechanisms, the authors note that paralogs can share enhancer sequences dating back to before duplication, which is indisputable, but then seem to imply either that each paralog's version of the enhancer has evolved in parallel under selection for high or low penetrance or that a single (or few) enhancer(s) is regulating other paralogs in trans (?). If I am interpreting correctly, more evidence should be provided for each of these claims. This section requires further clarification.

We improved the discussion related to this including that the differences between strains might be either in cis (enhancers) or trans (upstream activators) to the paralogs.

[Editors’ note: further revisions were suggested prior to acceptance, as described below.]

– Please make it clear throughout the intro and text that the manuscript consists of two components that are not yet clearly related.

We removed the first component, the allelic series study, at the suggestion of reviewer 2. We now focus on the strongest component, paralogous buffering.

– Discuss that the lack of understanding of how the various alleles affect expression or function of the gene's protein product, and the precise function of the gene in shaping craniofacial structures is not clear. Therefore, it is difficult to derive a mechanistic understanding of the observations and it remains to be investigated if the findings can be extrapolated from this study to other genes.

The various alleles have been removed.

– Please correct the text according to the issues identified by the two reviewers. The writing is often unclear: terms are used that are poorly defined or incorrectly applied; interpretations are freely mixed with presentations of results; paragraphs often have multiple ideas intermingled leading to a lack of clarity and repetition of ideas in different paragraphs. There are mistakes in the use of some references or the descriptions of some past work. Also descriptions of the precise experimental approach taken should be made clearer in some cases.

We have taken great care to define terms and apply them correctly. We saved all interpretations for the discussion and streamlined the text to remove repetitive ideas. We have double checked and corrected all references. We now include a precise description of the experimental approach as suggested.

Reviewer #2 (Recommendations for the authors):The revised manuscript is still very difficult to read and needs further improvementsThis work makes a substantial contribution to our understanding of how genetic background might function to modify the phenotype associated with a particular mutation. Three important take home lessons of this work are:1) Within populations there is existing (standing) variation in the expression levels of a set of paralogous genes;2) Control over the expression levels of these paralogues is heritable;3) Variations in the level of expression of paralogues can modify the expressivity and penetrance associated with a mutant allele of one of the paralogues. The authors use these observations to support a very interesting model they propose in which over evolution as a paralogue loses one of its functions, perhaps by acquiring diminished expression in a particular tissue, it may retain a vestigial capacity to support that function, which may be uncovered by boosting its expression in that tissue. The data presented appear consistent with the hypothesis that the levels of expression of each paralogue may be coordinately regulated, a model that could be readily addressed here but is not directly addressed in this study.

We thank this reviewer for appreciating this work. We emphasize the three important take home lessons described by this reviewer by explicitly listing them in the impact statement.

See further responses to specific critiques below.

The work is valuable. Its impact is dampened by four factors:1) The work makes two very different points that are not yet clearly related: the first portion of the manuscript deals with a purported relation between the strength of a mutant allele causing a developmental phenotype and the amount of variability in the structures that are formed abnormally, whereas the second portion of the manuscript deals with the idea that paralogue expression may mitigate or exaggerate the expression of the phenotype associated with a single mutation.

Per this reviewer’s suggestion, we removed the allelic series experiment and all interpretations purporting relation between the strength of a mutant *mef2ca* allele and the amount of variation.

2) The writing is often unclear: terms are used that are poorly defined or incorrectly applied; interpretations are freely mixed with presentations of results; paragraphs often have multiple ideas intermingled leading to a lack of clarity and repetition of ideas in different paragraphs.

We have carefully edited to ensure that terms are clearly defined and correctly used. We added a paragraph in which we defined some of the major terms, including robustness, compensation, and buffering. We are careful with our usage thereafter. We have taken great care to save interpretations for the discussion rather than mix with presentations of results.

3) There are outright mistakes in the use of some references or the descriptions of some past work.

We corrected these, thank you.

4) Descriptions of the precise experimental approach taken should be made clearer in some cases.

We have made clearer how different mutant lines were generated and maintained in the appropriate Results section.

Bailon-Zambrano et al. study factors that influence the expression of phenotype associated with a mutation. In one portion of the work they study a series of alleles of the zebrafish mef2ca gene, which is necessary for normal craniofacial development. They demonstrate that recessive putative loss-of-function mutant alleles of the mef2ca gene of zebrafish are associated with a range of expressivity. By focusing on one aspect of the mutant phenotype, the length of the symplectic cartilages that support the jaw, they find a correlation between the average strength of the phenotype of an allele (measured as reduction in length) and the extent of variability between mutant individuals that carry the allele. This is an interesting idea, but it is not fully explored or proven by their study. Because it is not clear how the various alleles affect expression or function of the gene's protein product, and because the precise function of the gene in shaping craniofacial structures is not clear, it is difficult to derive a mechanistic understanding of their observation and it is not possible to extrapolate from this study to other genes.

As suggested by this reviewer, we removed the first portion containing the allelic series from this study. The result is a streamlined body of work focusing on the paralogs.

The findings that associate individual mef2ca alleles with variability in the severity of the craniofacial phenotype are difficult to interpret. The authors order the "strength" of an allele based on the overall severity of the phenotype, but the molecular reason why the different alleles have different "strengths" is completely unclear. The mildest allele has a mutation affecting the initiation codon, and so presumably it produces a protein utilizing a different initiation point. However it is not clear why the deletion of almost the entire locus leads to an intermediate phenotype, whereas a mutation introducing a premature termination causes the most severe phenotype. Perhaps the severe allele produces a product that interferes with the activity of paralogous products or interacting proteins? Who knows? In addition, since we don't know the precise cellular functions regulated by the gene, it is impossible to generalize from this case and conclude, as the authors do, that there is a general correlation between severity of an allele and the variability in phenotype that results.

We removed the allelic series from this study.

Finally, this reviewer is concerned about this conclusion and generalizations that may be drawn from focus on a single quantifiable character, the symplectic cartilage. Perhaps there is always a fixed amount of variation in the length of this cartilage. As stronger alleles produce shorter cartilage pieces, similar absolute variations in length may appear to be of greater significance when affecting structures of shorter average length.

We removed the allelic series from this study. In the remaining phenotypic characterizations, we always include penetrance scoring of all *mef2ca* mutant-associated phenotypes to assay all *mef2ca* mutant-associated phenotypes.

We use the coefficient of variation to compare symplectic length variation between the high- and low-penetrance strains. We have confirmed with a statistician and coauthor (K. Colborn) that this value is appropriate for comparing variation in symplectic cartilage length because it uses a ratio of the variation and the mean, which allows you to compare variables of different scales or magnitude. Also, because the measurements are from a ratio scale, they are appropriate variables for estimating the coefficient of variation (for example, an ordinal or nominal variable would be inappropriate).

Finally, we also use the F-test to determine significant differences in variation.

In contrast with the first portion of the work, the second portion of the work makes truly valuable and mechanistic contributions to our understanding of how genetic background might modify the phenotypic expression of a particular mutation. The second portion investigates one mechanism that may contribute to the oft-observed phenomenon that a single mutation may be associated with different expressions of a phenotype in a manner that is dependent on the genetic background. The authors focus on loss-of-function of the mef2ca gene of zebrafish, which is needed for the normal development of several craniofacial structures. They find expression levels of mef2 paralogues may modify the phenotype associated with the mef2ca mutation. Three important take home lessons of this work are:1) Within populations there is existing (standing) variation in the expression levels of a set of paralogous genes;2) Control over the expression levels of these paralogues is heritable;3) Variations in the level of expression of paralogues can modify the expressivity and penetrance associated with a mutant allele of one of the paralogues. The authors use these observations to support a very interesting model they propose in which over evolution as a paralogue loses one of its functions, perhaps by acquiring diminished expression in a particular tissue, it may retain a vestigial capacity to support that function, which may be uncovered by boosting its expression in that tissue.

We are pleased that the reviewer thinks that “the work makes truly valuable and mechanistic contributions”

Major issues:1) I think the work is important, I would like to see it published, but I found the manuscript very difficult to read.

By removing the allelic series, we made the paper clearer and more concise. We have also clarified language and edited the manuscript according to this reviewer’s suggestions.

2) Perhaps one reason it is difficult to follow is that the work described here is composed of two very different studies, one describing the phenotypic effects of three different alleles and the other describing the effects of genetic background on the expression of one allele. This is reflected in the presence of two independent sections in the Introduction. Because we don't understand how the three different mutations affect generation of a protein product, or how they might affect the expression of paralogues (which are shown to be very important in the second half of this work), we cannot understand what a "strong" or "intermediate" allele is. IF the deletion completely removes gene function, then from a molecular point of view, it is a complete loss of function, and thus the premature termination mutation is a "special" type of allele that is more than simply completely loss of function. In any case, without insight into how these alleles work, I don't think it is fair to make generalizations about strengths of alleles and variability in the expression of a phenotype.

We agree and removed the allelic series component of the study.

On the other hand, there are clear statements in the Introduction that could really be used as the crux of an entire paper: "whether paralog expression variation underlies phenotypic variation has not been directly tested. Finally, whether variation paralogous buffering is subject to selection is unknown." Personally, I suggest focusing on the effects of paralogues to buffer phenotype – it makes for a crisp introduction and focused story.

As suggested, in this revised version we focus “on the effects of paralogues to buffer phenotype”.

3) There are assertions that I believe are incorrect.For example, lines 110-113: "Because mutations in none of the other four zebrafish paralogous mef2 genes have been reported, their function is unknown. These paralogs have human orthologs, MEF2A, MEF2B and MEF2D, that do not appear to be critical for craniofacial development46-48." Reference 46 and perhaps the others does NOT address the question of whether these genes are required for craniofacial development – ref 46 says mef2d fusion proteins are associated with cancer. These are somatic gain-of-function mutations and irrelevant to the matter at hand, which is complete whole animal loss of function.

We agree that only “complete whole animal loss of function” is relevant to the matter at hand and have revised this section accordingly.

Another example, lines 307-311: "Data showing that paralog transcriptional adaptation does not account for the phenotypic differences between the low- and high-penetrance strains34 do not rule out the possibility that selective breeding changed paralog expression between strains in mef2ca wild types." Reference 34 is the classic Force et al. paper and is not the correct reference – I think you meant to use reference 52.

We have corrected this – thank you.

Another example, lines 364-367: "We detected mef2d transcripts in the anterior arch population of 24 hpf wild-type cranial neural crest cells, and mef2d expression is significantly lower in the high-penetrance strain compared with the high-penetrance strain" I think you meant to compare high and low penetrance strains.

We have corrected this – thank you.

There are other examples of assertions that are a bit glib and that distract from the strength of the work.

We have carefully edited and revised the manuscript.

4) Terminology is used loosely or incorrectly.

We now include a section defining terms and have taken great care to use them correctly.

I find in the literature the use of between-individual as well as among-individual, and that between-individual is used more often. Personally I find between-individual easier to keep in my mind what the comparison being described is, but both are legit I guess.

We’ve opted to keep among-individual.

In many cases I believe it would be clearer to say that there are differences in expression levels among members of a population and that these levels are heritable – rather than saying "variability is heritable" (line 42 and elsewhere), which is a difficult concept to grasp precisely.

We removed this statement (line 42) and clarified elsewhere that the expression levels are heritable.

Lines 17-18: "Craniofacial disorders further increase this variability." "this variability" is ill-defined – perhaps it is more useful to speak of "range of variability"

We edited this statement for clarity.

Line 44: "Finish people" should be "Finnish people".

We have corrected this, thank you.

Lines 63-64: "Human genetic craniofacial disease phenotypes also appear more variable than normal human facial phenotypes." What is the actual comparison that is being made? I can't reconstruct this experiment.

We added a citation to support this claim.

Lines 70-71: "Patients heterozygous for mutations affecting the transcription factor encoding gene MEF2C show variable facial dysmorphologies19-21." This is relevant to the present study if you are describing variability associated with a single mutation – if that is true, make it explicit. If you mean that there are different alleles and these different alleles generate a range of phenotypes then it is not relevant to the discussion here.

We clarify in this section that multiple alleles are associated with *MEF2C* haploinsufficiency syndrome. Therefore, there is a need for a controlled model system that can be used to explore the variation that persists when the same allele is analyzed in different individuals.

Line 187 – not a "range of alleles", but a series of alleles.

We removed the allelic series study.

Line 225-227 and elsewhere – there are lots of places where alleles are described as variable – where the mutation or gene is conflated with the phenotype. "and the most severe mutant (mef2cab1086) allele was significantly more variable than the mildest mutant (mef2cab631) allele".

We removed the allelic series study and are careful to not conflate mutation or gene with the phenotype.

Line 266: "likely to be robust against null mutations" this is not clear.

We clarified the sentence.

The entire section beginning with line 284: "Closely related mef2 paralogs share similar expression dynamics" It is not clear what conclusions can be made or what conclusions you want to make about similar "dynamics". What one would like to measure is whether the paralogues are expressed in the same cells as the gene of interest and at a time when they are performing a function of interest. Since you have not explained when and where mef2ca functions, it is impossible to interpret the "dynamics" data in terms of relevant function. Please be explicit about what these experiments let you conclude and why.

We clarified this section to indicate when and where *mef2ca* functions and include relevant references. We then clarify that we used qPCR on whole heads to study the temporal dynamics of the paralogs. We then state that we use single-cell RNA sequencing to examine spatial expression of the paralogs.

Line 296-297: this paragraph is ended with a summary sentence: "Of note, previous work indicates that partially shared regulatory motifs are predictive of robustness66." Neither the meaning of this nor its relevance is clear to me. whose robustness? And in what context?

We deleted this statement.

5) Some things can just be made simpler:Lines 32-34: "We present a novel, mechanistic model for phenotypic variation where cryptically variable, vestigial paralog expression buffers development." I don't think the Reader will understand "cryptically variable" at this point of the paper (the Abstract).

We have removed this from the abstract.

Line 61 and elsewhere: "genetic mutations" could be simply "mutations".

We changed this as suggested.

6) Experiments and Models that need to made clearer:A critical series of experiments involves study of mutant alleles of the paralogues that are thought to be insufficient to have a dramatic effect on the craniofacial phenotype, but are thought to modify this phenotype in the context of mef2ca or other paralogue mutations. The authors at one point talk about this experiment as simply "removing" the paralogue (line 343). At that point, I had assumed the new mutations were being generated within the strain that was being studied – that is the easiest way to interpret "remove". Instead, paralogue mutations are induced elsewhere and bred into strains for study. The rationale for the experimental design is complicated and should be explicitly set forward. I would ask the authors to make clearer within the Results section how they generated the new mutations in the paralogues and how these were introduced into various strains and what precisely was examined. The entire exposition is a bit buried in Lines 336 – 338 paralogs. "For all paralog functional experiments, we introduced the relevant paralog mutation into the low-penetrance strain, then maintained by outcrossing to unselected AB." This is not a complete sentence. The details of this experiment are really important.

We clarified in the Results section how each paralog was mutagenized and maintained and provide details of the experiment in the relevant section. While we agree that a more elegant approach would be to perform the experiments in true breeding low-penetrance strains, this was not feasible due to time and space constraints (for example, rederiving new low-penetrance strain after crossing in the *mef2cb* allele). We are careful with our language to indicate that the *mef2cb* allele needed to be introduced by crossing into the low-penetrance background. Further we describe how after initially introducing the CRISPR/Cas9 lesions in the low-penetrance strain they were maintained outcrossing to an unselected AB background.

The other point that needs to be better explained and clarified is their model in Figure 9. The mechanistic implications of the illustration are vague, but to me there appears to be focus on the roles of enhancers – the figure seems to imply (i) that enhancers acquire mutations that diminish expression of the paralogue and (ii) that under selection the enhancer acquires mutations that result in increased expression of the gene. I would think rather that standing variation arises prior to the selection – and this might be illustrated in the figure. Nevertheless, it seems from the data that the expression levels of all the paralogues may be coordinately regulated – and this idea seems to supported in the Discussion. If all the paralogues are expressed at low levels in the high penetrance strain and at higher levels in the low penetrance strain, then it is more likely that variation is not being driven at the level of each paralogue's enhancer but rather by some factor working in trans. That idea is really missing from Figure 9 – and since it is a summary illustration of what the authors think, getting this Figure right is important. In truth, I would have liked to see a few crosses between strains to test the idea that a single modifier working in trans is responsible for all the effects – THAT would have been both relatively easy to test and killer interesting as a result. But without such an experiment, the authors need to be very clear about alternate possible interpretations.

We agree that getting this figure right is important. We completely revised our graphical model to summarize and illustrate the conclusions from this work.

Regarding crosses between strains: We previously published the hybridization experiment this reviewer suggests (Nichols et al. *Development,* 2016). Briefly, we crossed low- and high-penetrance individuals to each other. In this experiment: “We found that the F1 progeny had a mean phenotype between that of the parental lines, although the distribution of penetrance in the two F1 families differed slightly and both were skewed towards higher penetrance.” We further interpreted these data (and other data in the 2016 manuscript) to indicate that penetrance is inherited as a threshold character. In this model, inheritance of continuous variables results in an outwardly discontinuous phenotypic value. Our new work here indicates that the continuous variables we predicted are paralog expression levels, and the discontinuous phenotype is penetrance (a phenotype is present or not).

Finally, in that work our data supported a model where a few genetic factors control penetrance. In other words, it is oligogenic inheritance.

Reviewer #3 (Recommendations for the authors):The authors have extensively revised and added new data to their manuscript in response to the initial round of reviews. The language is generally more precise and concise, and sections lacking clarity are improved or have been removed. I especially appreciate that they now explain the breeding scheme by which the paralog mutant alleles were established and maintained, as the phenotype prevalences are now better in line with what the reader is led to predict.

Thank you for your comments which improved the manuscript.